# Zero-Cost Operation Scoring in Differentiable Architecture Search

## Abstract

Differentiable neural architecture search (NAS) has attracted significant attention in recent years due to its ability to quickly discover promising architectures of deep neural networks even in very large search spaces. Despite its success, many differentiable NAS methods lack robustness and may degenerate to trivial architectures with excessive parameter-free operations such as skip connections thus leading to inferior performance. In fact, selecting operations based on the magnitude of architectural parameters was recently proven to be fundamentally wrong, showcasing the need to rethink how *operation scoring* and selection occurs in differentiable NAS. To this end, we formalize and analyze a fundamental component of differentiable NAS: local "operation scoring" that occurs at each choice of operation. When comparing existing operation scoring functions, we find that existing methods can be viewed as inexact proxies for accuracy. We also find that existing methods perform poorly when analyzed empirically on NAS benchmarks. From this perspective, we introduce new training-free proxies to the context of differentiable NAS, and show that we can significantly speed up the search process while improving accuracy on multiple search spaces. We take inspiration from zero-cost proxies that were recently studied in the context of sample-based NAS but shown to degrade significantly for larger search spaces like DARTS. Our novel "perturbation-based zero-cost operation scoring" (Zero-Cost-PT) improves searching time and accuracy compared to the best available differentiable architecture search for many search space sizes, including very large ones. Specifically, we are able improve accuracy compared to the best current method (DARTS-PT) on the DARTS CNN search space while being over $40\times$ faster (total searching time 25 minutes on a single GPU). Our code is available at: https://github.com/avail-upon-acceptance.

## 1 Introduction

Since the recent dawn of deep learning, researchers have designed new architectures of neural networks on an unprecedented scale, with more efficient and accurate networks being proposed each year (Iandola et al., 2016; Howard et al., 2017; Tan & Le, 2019; 2021). However, the manual design of better DNN architectures has proven quite challenging, requiring extensive domain knowledge and persistent trial-and-error in search of the optimal hyperparameters (Sandler et al., 2018; Tan & Le, 2021). Recently this process has been successfully aided through automated methods, especially neural architecture search (NAS) which can be found behind many of the state-of-the-art deep neural networks (Real et al., 2019; Wu et al., 2019; Cai et al., 2020; Moons et al., 2020). However, one of the biggest problems in NAS is the computational cost – even training a single deep network can require enormous computational resources and many NAS methods need to train tens, if not hundreds, of networks in order to converge to a good architecture (Real et al., 2019; Luo et al., 2018; Dudziak et al., 2020). A related problem concerns the search space size – a larger search space would typically contain better architectures, but requires longer searching time (Real et al., 2019).

Differentiable architecture search (DARTS) was first proposed to tackle those challenges, showcasing promising results when searching for a network in a set of over $10^{18}$ possible variations (Liu et al., 2019). Unfortunately, DARTS has significant robustness issues as demonstrated through many recent works (Zela et al., 2020a; Shu et al., 2020; Yu et al., 2020). It also requires careful selection of hyperparameters which makes it somewhat difficult to adapt to a new task. Recently, Wang et al. (2021) showed that operation selection in DARTS based on the magnitude of architectural parameters ($\alpha$) is fundamentally wrong, and will always simply select skip connections over other more meaningful operations. They proposed an alternative operation selection method based on *perturbation*, where the importance of an operation is determined by the decrease of the supernet's validation accuracy

when it is removed. Then the most important operations are selected by exhaustively comparing them with other alternatives on each single edge of the supernet, until the final architecture is found.

In a parallel line of work that aims to speed up NAS, *proxies* are often used instead of training accuracy to obtain an indication of performance quickly without performing expensive full training for each searched model. Conventional proxies typically consist of a reduced form of training with fewer epochs, less data or a smaller DNN architecture (Zhou et al., 2020). Most recently, *zero-cost proxies*, which are extreme types of NAS proxies that do not require any training, have gained interest and are shown to empirically outperform conventional training-based proxies and deliver outstanding results on common NAS benchmarks (Abdelfattah et al., 2021; Mellor et al., 2021). However, their efficient usage on a large search spaces, typical for differentiable NAS, has been shown to be more challenging and thus remains an open problem (Mellor et al., 2021).

The objective of our paper is to shed some light onto the implicit proxies that are used for operation scoring in differentiable NAS, and to propose new proxies in this setting that have the potential of improving both search speed and quality. We decompose differentiable NAS into its two constituent parts: (1) supernet training and (2) operation scoring. We focus on the second component and we formalize the concept of "operation scoring" that happens during local operation selection at each edge in a supernet. Through this lens, we are able to empirically compare the efficacy of existing differentiable NAS operation scoring functions. We find that existing methods act as a proxy to accuracy and perform quite poorly on common NAS benchmarks, consequently, we propose new operation scoring functions based on zero-cost proxies that outperform existing methods in both search speed and accuracy. Our main contributions are:

1. Formalize "operation scoring" in differentiable NAS and perform a first-of-its-kind analysis of the implicit proxies that are present in existing differentiable NAS methods.
2. Propose, evaluate and compare perturbation-based zero-cost operation scoring (Zero-Cost-PT) for differentiable NAS building upon recent work on training-free NAS proxies.
3. Perform a thorough empirical evaluation of Zero-Cost-PT in six search spaces and 3 datasets including DARTS, DARTS subspaces S1-S4 and NAS-Bench-201.

## 2 RELATED WORK

**Classic NAS and Proxies.** Zoph & Lee were among the first to propose an automated method to search neural network architectures, using a reinforcement learning agent to maximize rewards coming from training different models (Zoph & Le, 2017). Since then, a number of alternative approaches have been proposed in order to reduce the significant cost introduced by training each proposed model. In general, reduced training can be found in many NAS works (Pham et al., 2018; Zhou et al., 2020), and different proxies have been proposed, e.g. searching for a model on a smaller dataset and then transfer the architecture to the larger target dataset (Real et al., 2019; Mehrotra et al., 2021) or incorporating a predictor into the search process (Wei et al., 2020; Dudziak et al., 2020; Wu et al., 2021; Wen et al., 2019).

**Zero-cost Proxies.** Very recently, zero-cost proxies (Mellor et al., 2021; Abdelfattah et al., 2021) for NAS emerged from pruning-at-initialisation literature (Tanaka et al., 2020; Wang et al., 2020; Lee et al., 2019; Turner et al., 2020). Such proxies can be formulated as architecture scoring functions $S(A)$ that evaluate the potential or "saliency" of a given architecture $A$ in achieving accuracy at initialization, without the expensive training process. In this paper, we adopt the recently proposed zero-cost proxies (Abdelfattah et al., 2021; Mellor et al., 2021), namely `grad_norm`, `snip`, `grasp`, `synflow`, `fisher` and `nwot`. Those metrics either aggregate the saliency of model parameters to compute the score of an architecture (Abdelfattah et al., 2021), or use the overlapping of activations between different samples within a minibatch of data as a performance indicator (Mellor et al., 2021). In a similar vein, Chen et al. (2021) proposed the use of training-free scoring for operations based on the neural tangent kernel (Jacot et al., 2021) and number of linear regions in a DNN; the operations with the lowest score are *pruned* from the supernet iteratively until a subnetwork is found.

**Differentiable NAS and Operation Perturbation.** Liu et al. first proposed to search for a neural network's architecture by parameterizing it with continuous values (called architectural parameters $\alpha$) in a differentiable way. Their method constructs a *supernet*, i.e., a superposition of all networks in the search space, and optimizes the architectural parameters ($\alpha$) together with supernet weights ($w$). The final architecture is extracted from the supernet by preserving operations with the largest $\alpha$. Despite the significant reduction in searching time, the stability and generalizability of DARTS have been challenged, e.g., it may produce trivial models dominated by skip connections (Zela et al.,

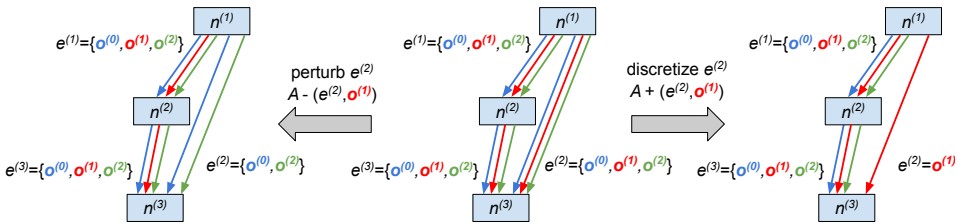

Figure 1: Visualization of perturbation and discretization of an edge in a supernet. Middle: a supernet is composed of three edges $\{e^{(i)}\}_{i=1,2,3}$, each consisting of three possible operations $\{o^{(i)}\}_{i=1,2,3}$ which are applied in parallel to the same input. Left: edge $e^{(2)}$ is perturbed by removing $o^{(1)}$ from the set of candidate operations assigned to this edge. Right: the same edge $e^{(2)}$ is discretized with operation $o^{(1)}$ by removing all other candidate operations leaving $o^{(1)}$ as the only choice left.

2020a). SDARTS (Chen & Hsieh, 2020) proposed to overcome such issues by smoothing the loss landscape, while SGAS (Li et al., 2020) considered a greedy algorithm to select and prune operations sequentially. The recent DARTS-PT (Wang et al., 2021) proposed a perturbation-based operation selection strategy, showing promising results on DARTS space. In DARTS-PT operations are no longer selected by optimizing architectural parameters ($\alpha$), but via a scoring function evaluating the impact on supernet's validation accuracy when the operation is removed.

## 3 RETHINKING OPERATION SCORING IN DIFFERENTIABLE NAS

In the context of differentiable NAS, a supernet would contain multiple candidate operations on each edge as shown in Figure 1. Operation scoring functions assign a score to rank operations and select the best one. In this section, we empirically quantify the effectiveness of existing operation scoring methods in differentiable NAS, with a specific focus on DARTS (Liu et al., 2019) and the recently-proposed DARTS-PT (Wang et al., 2021). Concretely, we view these scoring functions as proxies for final subnetwork accuracies and we evaluate them on that basis to quantify how well these functions perform. We challenge many assumptions made in previous work and show that we can outperform existing methods with lightweight alternatives.

### 3.1 OPERATION SCORING PRELIMINARIES

For a supernet $A$ we want to be able to start *discretizing* edges in order to derive a subnetwork. When discretizing we replace an edge composed of multiple candidate operations and their respective (optional) architectural parameters $\alpha$ with only one operation selected from the candidates. We will denote the process of discretization of an edge $e$ with operation $o$, given a model $A$, as: $A + (e, o)$. Analogously, the *perturbation* of a supernet $A$ by removing an operation $o$ from an edge $e$ will be denoted as $A - (e, o)$. Figure 1 illustrates discretization and perturbation. Furthermore, we will use $\mathcal{A}$, $\mathcal{E}$ and $\mathcal{O}$ to refer to the set of all possible network architectures, edges in the supernet and candidate operations, respectively – extra details about notation can be found in the Appendix A.1.

NAS can then be performed by iterative discretization of edges in the supernet, yielding in the process a sequence of partially discretized architectures: $A_0, A_1, ..., A_{|\mathcal{E}|}$, where $A_0$ is the original supernet, $A_{|\mathcal{E}|}$ is the final fully-discretized subnetwork (result of NAS), and $A_t$ is $A_{t-1}$ after discretizing a next edge, i.e., $A_t = A_{t-1} + (e_t, o_t)$ where $t$ is an iteration counter. The problem of finding the sequence of $(e_t, o_t)$ that maximizes performance of the resulting network $A_{\mathcal{E}}$ has an optimal substructure and can be reduced to the problem of finding the optimal policy $\pi : \mathcal{A} \times \mathcal{E} \rightarrow \mathcal{O}$ that is used to decide on an operation to assign to an edge at each iteration, given current model (state). This policy function is defined by means of an analogous scoring function $f : \mathcal{A} \times \mathcal{E} \times \mathcal{O} \rightarrow \mathbb{R}$, that assigns scores to the possible values of the policy function, and the policy is then defined as $\arg\max$ or $\arg\min$ over $f$, depending on the type of scores produced by $f$. [1]

We begin by defining the optimal scoring function that we will later use to assess quality of different, practical approaches. For a given partially-discretized model $A_t$, let us denote the set of all possible fully-discretized networks that can be obtained from $A_t$ after a next edge $e$ is discretized with an operation $o$ as $\mathcal{A}_{t,e,o}$. Our optimal scoring function can then be defined as:

$$\pi_{\text{best-acc}}(A_t, e) = \arg\max_{o \in \mathcal{O}_e} \max_{A_{|\mathcal{E}|} \in \mathcal{A}_{t,e,o}} V^*(A_{|\mathcal{E}|}) \tag{1}$$

---

[1] Since a scoring function clearly defines a relevant policy function, we will sometimes talk about a scoring function even though the context might be directly related to a policy function – in those cases it should be understood as the policy function that follows from the relevant scoring function (and vice versa).

where $V^*$ is validation accuracy of a network after full training (we will use $V$ to denote validation accuracy without training). It is easy to see that this policy meets Bellman's principle of optimality (Bellman, 1957) – in fact its definition follows directly from it – and therefore is the optimal solution to our problem. However, it might be more practical to consider the expected achievable accuracy when an operation is selected, instead of the best. Therefore we also define the function $\pi_{\text{avg}}$:

$$\pi_{\text{avg-acc}}(A_t, e) = \arg \max_{o \in \mathcal{O}_e} \mathbb{E}_{A_{|\mathcal{E}|} \in \mathcal{A}_{t,e,o}} V^*(A_{|\mathcal{E}|}) \tag{2}$$

In practice, we are unable to use either $\pi_{\text{best-acc}}$ or $\pi_{\text{avg-acc}}$ since we would need to have the final validation accuracy $V^*$ of all the networks in the search space. There have been many attempts at finding approximate operation scoring functions, in the following we consider the following practical alternatives from DARTS (Liu et al., 2019) and DARTS-PT (Wang et al., 2021):

$$\pi_{\text{darts}}(A_t, e) = \arg \max_{o \in \mathcal{O}_e} \alpha_{e,o} \tag{3}$$

$$\pi_{\text{disc-acc}}(A_t, e) = \arg \max_{o \in \mathcal{O}_e} V^*(A_t + (e, o)), \quad \pi_{\text{darts-pt}}(A_t, e) = \arg \min_{o \in \mathcal{O}_e} V(A_t - (e, o)) \tag{4}$$

where $\alpha_{e,o}$ is the architectural parameter assigned to operation $o$ on edge $e$ as presented in DARTS (Liu et al., 2019). $\pi_{\text{disc-acc}}$ uses accuracy of a supernet after an operation $o$ is assigned to an edge $e$ – this is referred to as "discretization accuracy" in DARTS-PT and is assumed to be a good operation scoring function (Wang et al., 2021), most intuitively, it could approximate $f_{\text{avg-acc}}$. $\pi_{\text{darts-pt}}$ is the perturbation-based approach used by DARTS-PT – it is presented as a practical and lightweight alternative to $\pi_{\text{disc-acc}}$ (Wang et al., 2021).

**Zero-Cost Operation Scoring.** We argue that the scoring functions (3) and (4) are merely proxies for the best achievable accuracy (Equation 1). As such, we see an opportunity to use a new class of training-free proxies that are very fast to compute and have been shown to work well within NAS, albeit not in differentiable NAS, nor within large search spaces. We present the following scoring functions that use a zero-cost proxy $S$ instead of validation accuracy when discretizing an edge or perturbing an operation. Note that the supernet is randomly-initialized and untrained in this case.

$$\pi_{\text{disc-zc}}(A_t, e) = \arg \max_{o \in \mathcal{O}_e} S(A_t + (e, o)), \quad \pi_{\text{zc-pt}}(A_t, e) = \arg \min_{o \in \mathcal{O}_e} S(A_t - (e, o)) \tag{5}$$

Note that TE-NAS (Chen et al., 2021) also uses training-free scoring of operations, however, they use different scoring metrics to *prune* operations from a supernet as opposed to discretizing or perturbing operations as we show above. We include a comparison to TE-NAS throughout our paper.

### 3.2 Empirical Evaluation of Operation Scoring Methods

In this subsection we investigate the performance of different operation scoring methods. Because we want to compare with the optimal $f_{\text{best-acc}}$ and $f_{\text{avg-acc}}$, we conduct experiments on NAS-Bench-201 which contains the validation accuracy for all 15,625 subnetworks in the search space (Dong & Yang, 2020). We conduct our investigation in two settings, *initial* and *progressive*. The first setting compares operation scoring functions while making their first decision (iteration 0) during NAS. The second (*progressive*) setting takes into account retraining that occurs between iterations for some of the differentiable NAS algorithms that we consider like darts-pt (Wang et al., 2021). ==Throughout this section, whenever we discuss a method based on zero-cost scoring, we use `naswot` metric== (Mellor et al., 2021).

#### 3.2.1 Initial Operation Scoring

For the supernet $A_0$ we compute the operation scores for all operations on all edges, at the first iteration (iteration 0) of NAS, that is, $f(A_0, e, o) \ \forall \ e \in \mathcal{E}, \ o \in \mathcal{O}_e$. In our first experiment, we collect the scores produced by different scoring methods, per operation, per edge, then compute the Spearman rank correlation for operations on each edge, and finally average the rank correlation coefficient over all edges (details of our experiments and illustrative examples are provided in Appendix A.3). The resulting averaged rank correlation is indicative of how well an operation scoring method would do when making the first discretization decision, relative to a perfect "oracle" search. We plot the rank correlation coefficients in Figure 2a, showing many surprising findings. First, disc-acc is inversely correlated to best-acc. This refutes the claim in the DARTS-PT paper that disc-acc is a reasonable operation score (Wang et al., 2021) – these findings are aligned with prior work that has already shown that the supernet accuracy is unrelated to the final subnetwork accuracy (Li et al., 2020). Second, the darts-pt score does not track disc-acc, in fact, it is inversely-correlated to it as well, meaning that the darts-pt score is not a good approximation of disc-acc. However, darts-pt is weakly-correlated to the "oracle" best-acc and avg-acc scores which supports (empirically) why it works well. Third, our zc-pt

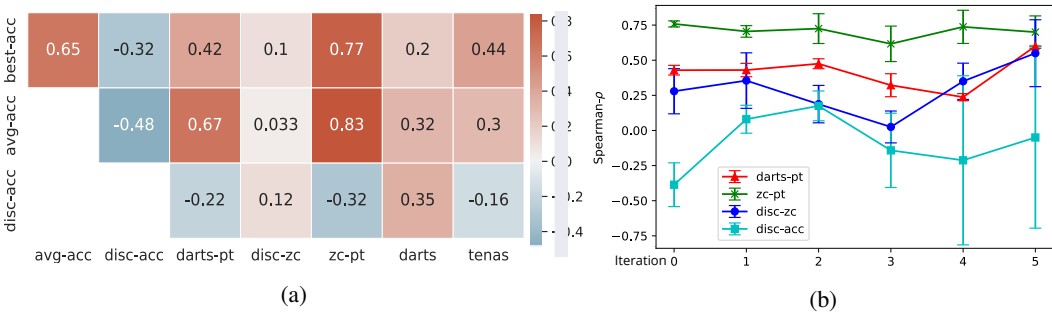

(a)  (b)

Figure 2: (a) Spearman's rank correlation coefficient of different operation scoring metrics with each other at the first iteration of NAS. (b) Rank correlation coefficient of different operation scoring functions vs. best-acc when invoked iteratively for each edge. In iteration $i$, only edge $i$ is discretized then all scores for all operations on the remaining edges is computed and correlated against best-acc.

Table 1: Model selected based on maximizing each operation strength independently.

| | best-acc | avg-acc | disc-acc | darts-pt | zc-pt | disc-zc | darts | tenas |
|---|---|---|---|---|---|---|---|---|
| Avg. Error[1][%] | 5.63 | 6.24 | 13.55 | 19.43 | 5.81 | 22.96 | 45.7 | 7.19 |
| Rank in NAS-Bench-201 | 1 | 166 | 12,744 | 13,770 | 14 | 14,274 | 15,231 | 1,817 |

[1] Computed as the average of all available seeds for the selected model in NAS-Bench-201 CIFAR-10 dataset.

is strongly-correlated with both the best-acc and avg-acc metrics, indicating that there could be huge promise when using this scoring function within NAS. Note that disc-zc, like disc-acc is inversely correlated with the oracle score suggesting that *perturbation* is generally a better scoring paradigm than *discretization*. Fourth, tenas (Chen et al., 2021), which also utilizes training-free operation scoring, performs fairly well, with Spearman-$\rho$ =0.44, but still falls short of the performance of zc-pt ($\rho$ =0.77). Finally, the original darts $\alpha$ score is weakly and inversely correlated with the oracle scores, further supporting arguments in prior work that this is not an effective operation scoring method.

In Table 1 , we show the discovered NAS-Bench-201 architecture when applying the seven scoring functions (Eq. (1) – (5)) for operation selection on all edges. As expected, best-acc chooses the best subnetwork, while avg-acc selects a very good model but not the best one, likely due to the large variance of accuracies in NAS-Bench-201. zc-pt selected one of the top models in NAS-Bench-201 as expected from the strong correlation with the oracle best-acc function; while tenas selected a good model, in the top 15% of the NAS-Bench-201 dataset, commensurate with the average correlation shown in Figure 2a. The remaining operation scoring functions failed to produce a good model in this experiment, suggesting that these metrics do not make a good initial choice of operations at iteration 0 of differentiable NAS. While this signals a major weakness of those differentiable NAS proxies, it's worthwhile to further analyze these methods in the *progressive* setting which would show what happens in later iterations of NAS. To further investigate the initial behaviour of different scoring methods, as well as their effects on NAS performance, we run similar experiments on NAS-Bench-1shot1 search space, the results can be found in Appendix A.8.

### 3.2.2 PROGRESSIVE OPERATION SCORING

Until now, we have only investigated the performance of operation scoring functions in the first iteration of NAS. This approach is relevant for methods like DARTS, where operation scoring function $f$ does not depend on $A_t$ in any way (only $A_0$), but is not truly representative of other methods that work iteratively. Because of that, we extend our analysis to investigate what happens in later iterations of NAS. In order to do that, we calculate the correlation of scoring functions in the *progressive* setting by performing the following steps: (1) score operations on all undiscretized edges, (2) discretize edge $i$, (3) retrain for 5 epochs (darts-pt and disc-acc only), (4) increment $i$ and repeat from step 1 until all edges are discretized. At each iteration $i$, we calculate the scores for the operations on all remaining undiscretized edges and compute their Spearman rank correlation coefficients (Spearman-$\rho$) with respect to best-acc. This is plotted in Figure 2b, averaged over four seeds.

Our results confirm many of our *initial* (iteration-0) analysis. zc-pt continues to be the best operation scoring function, and darts-pt is the second-best, improving in correlation from 0.4 to 0.6 between the first and last iterations, indeed showing that retraining and/or progressive discretization helps. However, disc-acc continues to be unrepresentative of operation strength even when used in the iterative setting. This is not what we expected, especially in the very last iteration when disc-acc is supposed to match a subnetwork exactly. As Figure 2b shows, the variance in the last iteration is quite large – we believe this happens because we do not train to convergence every time we discretize

an edge, and instead we only train for 5 epochs. Our progressive analysis provided further empirical evidence that supernet discretization accuracy should not be used as a proxy for subnetwork accuracy, contradicting Wang et al. (2021). However, we have confirmed that darts-pt does in fact improve when retraining is performed between NAS iterations, but could still be improved upon with zc-pt – it performed exceptionally well as a proxy for accuracy on NAS-Bench-201, and has the potential to make differentiable NAS both much faster and of higher accuracy.

## 4 ZERO-COST-PT NEURAL ARCHITECTURE SEARCH

In this section, we propose a NAS algorithm called Zero-Cost-PT based on zero-cost perturbation, and perform ablation studies to find the best set of heuristics for our search methodology.

### 4.1 ARCHITECTURE SEARCH WITH ZERO-COST PROXIES

Our algorithm contains two stages: *architecture proposal* and *validation*. It begins with an untrained supernet $A_0$ which contains a set of edges $\mathcal{E}$, the number of proposal iterations $N$, and the number of validation iterations $V$. In each proposal iteration $i$, we discretize the supernet $A_0$ based on our proposed zero-cost-based perturbation function $f_{\text{zc-pt}}$ that achieved promising results in the previous section. After all edges have been discretized, the final architecture is added to the set of candidates and we begin the process again for $i+1$ starting with the original $A_0$. After $N$ candidate architectures have been constructed, the validation stage begins. We score the candidate architectures again using a selected zero-cost metric (the same which is used in $f_{\text{zc-pt}}$), but this time computing their end-to-end score rather than using the perturbation paradigm. We calculate the zero-cost metric for each subnetwork using $V$ different minibatches of data. The final architecture is the one that achieves the best total score during the validation stage. The full algorithm is outlined as Algorithm 1 in Appendix A.2. Our algorithm contains four main hyperparameters: $N$, $V$, ordering of edges to follow when discretizing, and the zero-cost metric to use ($S$). In the following we present detailed ablations to decide on the best possible configuration of these.

### 4.2 ABLATION STUDY ON NAS-BENCH-201

We conduct ablations of the proposed Zero-Cost-PT approach on NAS-Bench-201 (Dong & Yang, 2020). NAS-Bench-201 constructed a unified cell-based search space, where each architecture has been trained on three different datasets, CIFAR-10, CIFAR-100 and ImageNet-16-120[2]. In our experiments, we take a randomly initialised supernet for this search space and apply our Zero-Cost-PT algorithm to search for architectures without any training. We run the search with four different random seeds (0, 1, 2, 3) and report the average and standard deviation of the test errors of the obtained architectures. All searches are performed on CIFAR-10, and obtained architectures are then additionally evaluated on the other two datasets.

**Different Zero-cost Metrics.** Since our focus is to understand how the existing zero-cost metrics can be successfully applied to a large-space NAS, we begin our investigation by analysis how different metrics behave when used in the proposed combination with perturbation-based search. In particular, we consider the following metrics that have been proposed in recent zero-cost NAS literature (Abdelfattah et al., 2021; Mellor et al., 2021): `grad_norm` (Abdelfattah et al., 2021), `snip` (Lee et al., 2019), `grasp` (Wang et al., 2020), `synflow` (Tanaka et al., 2020), `fisher` (Theis et al., 2018) and `nwot` (Mellor et al., 2021). Table 2 compares the average test errors of architectures selected by different proxies on NAS-Bench-201. We see that `nwot` and `synflow` perform considerably better across the three datasets than the others, where `nwot` offers

Table 2: Comparison in test error (%) with the state-of-the-art perturbation-based and zero-cost NAS on NAS-Bench-201 (Best in red, 2nd best in blue. Same for all following tables).

| Method | CIFAR-10 | CIFAR-100 | ImageNet-16 |
|---|---|---|---|
| **Zero-Cost-PT** with different proxies (Section 4.2) | | | |
| fisher | $10.64_{\pm 1.27}$ | $38.48_{\pm 1.96}$ | $82.85_{\pm 12.63}$ |
| grad_norm | $10.55_{\pm 1.11}$ | $38.43_{\pm 2.10}$ | $80.71_{\pm 12.10}$ |
| grasp | $9.81_{\pm 3.42}$ | $36.52_{\pm 6.33}$ | $64.27_{\pm 8.82}$ |
| snip | $8.32_{\pm 2.02}$ | $34.00_{\pm 4.03}$ | $65.35_{\pm 11.04}$ |
| synflow[1] | $6.24_{\pm 0.00}$ | $28.89_{\pm 0.00}$ | $58.56_{\pm 0.00}$ |
| nwot | $\mathbf{5.97}_{\pm 0.17}$ | $\mathbf{27.47}_{\pm 0.28}$ | $\mathbf{53.82}_{\pm 0.77}$ |
| **Baselines and SOTA approaches** (Section 5.1) | | | |
| Random | $13.39_{\pm 13.28}$ | $39.17_{\pm 12.58}$ | $66.87_{\pm 9.66}$ |
| DARTS | $45.70_{\pm 0.00}$ | $84.39_{\pm 0.00}$ | $83.68_{\pm 0.00}$ |
| DARTS-PT [1] | $11.89_{\pm 0.00}$ | $45.72_{\pm 6.26}$ | $69.60_{\pm 4.40}$ |
| DARTS-PT (fix $\alpha$) [2] | $6.20_{\pm 0.00}$ | $34.03_{\pm 2.24}$ | $61.36_{\pm 1.91}$ |
| NASWOT(synflow) [3] | $6.54_{\pm 0.62}$ | $29.53_{\pm 2.13}$ | $58.22_{\pm 4.18}$ |
| NASWOT(nwot) [3] | $7.04_{\pm 0.80}$ | $29.97_{\pm 1.16}$ | $55.57_{\pm 2.07}$ |
| TE-NAS | $6.10_{\pm 0.47}$ | $28.76_{\pm 0.56}$ | $57.62_{\pm 0.46}$ |

[1] Only 1 model was selected across all 4 seeds.
[2] Results on CIFAR-10 taken from (Wang et al., 2021). Results on other datasets computed using official code in (Wang et al., 2021) across 4 seeds.
[3] Using N=1000 for both proxies and averaged over 500 runs as in (Mellor et al., 2021).

---
[2] We use the three random seeds available in NAS-Bench-201: 777, 888, 999.

Table 3: Test error (%) of Zero-Cost-PT when using different search orders on NAS-Bench-201.

| Search Order[1] | # of Perturbations[2] | CIFAR-10 | CIFAR-100 | ImageNet-16 |
|---|---|---|---|---|
| fixed | $\|\mathcal{O}\|\|\mathcal{E}\|$ | $5.98_{\pm 0.50}$ | $27.60_{\pm 1.63}$ | $54.23_{\pm 0.93}$ |
| global-op-iter | $\frac{1}{2}\|\mathcal{O}\|\|\mathcal{E}\|(\|\mathcal{E}\|+1)$ | $\mathbf{5.69}_{\pm 0.19}$ | $\mathbf{26.80}_{\pm 0.51}$ | $\mathbf{53.64}_{\pm 0.40}$ |
| global-op-once | $2\|\mathcal{O}\|\|\mathcal{E}\|-\|\mathcal{O}\|$ | $6.30_{\pm 0.57}$ | $28.96_{\pm 1.66}$ | $55.04_{\pm 1.47}$ |
| global-edge-iter | $\frac{1}{2}\|\mathcal{O}\|\|\mathcal{E}\|(\|\mathcal{E}\|+1)$ | $6.23_{\pm 0.45}$ | $28.42_{\pm 0.59}$ | $54.39_{\pm 0.47}$ |
| global-edge-once | $2\|\mathcal{O}\|\|\mathcal{E}\|-\|\mathcal{O}\|$ | $6.30_{\pm 0.57}$ | $28.96_{\pm 1.66}$ | $55.04_{\pm 1.47}$ |
| random | $\|\mathcal{O}\|\|\mathcal{E}\|$ | $5.97_{\pm 0.17}$ | $27.47_{\pm 0.28}$ | $53.82_{\pm 0.77}$ |

[1] All methods use `nwot` metric, N=10 architecture proposal iterations and V=100 validation iteration.
[2] Number of perturbations per search iteration.

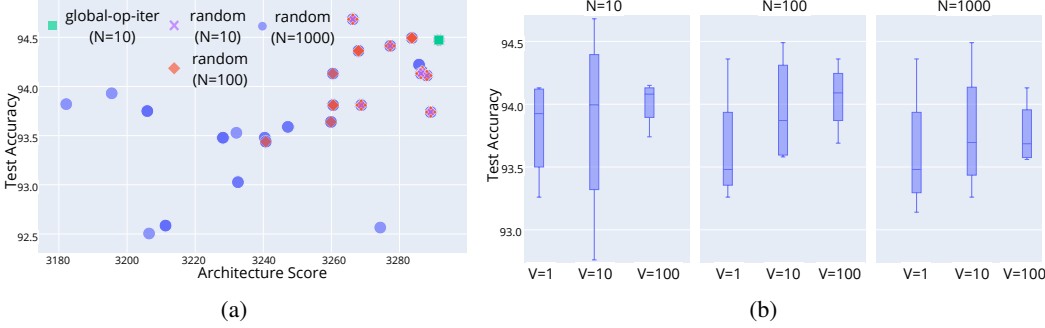

Figure 3: (a) Accuracy vs. score of architectures discovered on CIFAR-10 by Zero-Cost-PT with different N. (b) Accuracy distribution of discovered architectures with different N and V.

around 0.27% improvement over `synflow`. However, even the worst performing `fisher` and naive `grad_norm` outperform the state-of-the-art DARTS-PT on this benchmark (see Table 2). This confirms that the zero-cost metrics, when combined with the perturbation-based NAS paradigms as in Zero-Cost-PT, could become promising proxies to the actual trained accuracy. We also observed that the ranking of those metrics are quite stable on the three datasets (descending order in terms of error as in Table 2), indicating that architectures discovered by our Zero-Cost-PT have good transferability across datasets. `nwot` consistently performs best, reducing test errors on all three datasets by a considerable margin.

**Edge Discretization Order.** We also study how different edge discretization order may impact the performance of our Zero-Cost-PT approach. We consider the following edge discretization orders:

- `fixed`: discretizes the edges in a fixed order, where in our experiments we discretize from the input towards the output of the cell structure;
- `random`: discretizes the edges in a random order;
- `global-op-iter`: iteratively evaluates $S(A-(e,o))$ for all operations on all edges in $\mathcal{E}$, selects the edge $e$ containing the operation $o^*$ with globally best score. Discretizes $e$ with $o^*$, then repeats to decide on the next edge (re-evaluating scores) until all edges have been discretized;
- `global-op-once`: only evaluates $S(A-(e,o))$ for all operations once to obtain a ranking order of the operations and decide the edge order upfront based on it, then starts following the algorithm as usual, calculating scores of operations at each edge iteratively;
- `global-edge-iter`: similar to `global-op-iter` but iteratively selects edge $e$ from $\mathcal{E}$ based on the average score of all operations on each edge;
- `global-edge-once`: similar to `global-op-once` but uses the average score of operations on edges to obtain the edge discretization order.

In this experiments we run N=10 architecture proposal iterations and V=100 validation iterations for all variants, using `naswot` metric. Table. 3 shows the performance of the approaches. We see that the `global-op-iter` consistently performs best across all three datasets, since it iteratively explores the search space of remaining operations, while greedily selecting the current best. However, it comes with a higher cost than `fixed` or `random`, since we need to perform $\frac{1}{2}\|\mathcal{O}\|\|\mathcal{E}\|(\|\mathcal{E}\|+1)$ perturbations in total, while the latter require $\|\mathcal{O}\|\|\mathcal{E}\|$. On the other hand, we see that the performance of `global-op-once` is inferior since it determines the order of perturbation by assessing the importance of operations once for all at the beginning, which may not be appropriate as discretization continues. Note that when discretizing an edge according to the obtained order, `global-op-once` still need to perturb the $\|\mathcal{O}\|$ operations on each remaining edge. We observe similar behaviour in `global-edge-iter` and `global-edge-once`, both of which use the average importance of operations on edges to decide search order, leading to suboptimal performance. It is also worth

pointing out that `fixed` performs relatively well comparing to the other variants, offering comparable performance with `random`. This shows that Zero-Cost-PT is generally robust to the edge discretization order. For simplicity, in the following experiments we use `random` order with a moderate setting in architecture proposal iterations (N=10) to balance exploration and exploitation during search, while maintaining the efficiency of Zero-Cost-PT.

**Proposal vs. Validation.** We study the impact of different architecture proposal iterations N and validation iterations V when Zero-Cost-PT uses `random` as the search order and `nwot` metric. Intuitively, larger N leads to more architecture candidates being found, while V indicates the amount of data used to rank the search candidates. As shown in Figure 3a, we see larger N does lead to more architectures discovered, but not proportional to the value of N on NAS-Bench-201 space. For N=100 we discover 27.8 distinct architectures on average, but when increased to N=1000 the number only roughly

Table 4: Comparison with the state-of-the-art differentiable NAS methods on the DARTS CNN search space (CIFAR-10).

| Method | Test Error (%) | | Params (M) | Cost[2] |
|---|---|---|---|---|
| | Avg. | Best | | |
| DARTS | $3.00_{\pm 0.14}$ | - | 3.3 | 0.4 |
| SDARTS-RS | $2.67_{\pm 0.03}$ | - | 3.4 | 0.4 |
| SGAS | $2.66_{\pm 0.24}$ | - | 3.7 | 0.25 |
| DARTS-PT | $\mathbf{2.61}_{\pm 0.08}$ | 2.48 | 3.0 | 0.8 |
| DARTS-PT$_{+none}$[1] | $2.73_{\pm 0.13}$ | 2.67 | 3.2 | 0.8 |
| TE-NAS | $2.63_{\pm 0.064}$ | - | 3.8 | 0.05 |
| Zero-Cost-PT$_{random}$ | $2.64_{\pm 0.16}$ | **2.43** | 4.7 | **0.018** |
| Zero-Cost-PT$_{global-op-iter}$ | $2.62_{\pm 0.09}$ | 2.49 | 4.6 | 0.17 |

[1] Results obtained by re-enabling `none` operation in DARTS-PT (Wang et al., 2021).
[2] In GPU days. Cost of existing approaches taken from (Wang et al., 2021). Cost of Zero-Cost-PT measured on a single 2080Ti GPU.

doubles. We also see that even with N=10, Zero-Cost-PT$_{random}$ can already discover top models in the space, demonstrating desirable balance between search quality and efficiency. On the other hand, as shown in Figure 3b, larger V tends to reduce the performance variance, especially for smaller N. This is also expected as more validation iterations could stabilise the ranking of selected architecture candidates, helping Zero-Cost-PT to retain the most promising ones with a manageable overhead of V minibatches. We provide additional ablations regarding the effects of N and V conducted on the DARTS CNN space in Appendix A.7. We also revisit our comparison between `zc-pt` and `zc-disc` from Section 3.2 by comparing our Zero-Cost-PT when run with either scoring method on NAS-Bench-201 in Appendix A.6.3.

# 5 RESULTS

In this section we perform extensive empirical comparisons of Zero-Cost-PT with the state-of-the-art differentiable and zero-cost NAS algorithms on a number of search spaces, including NAS-Bench-201 (Dong & Yang, 2020), DARTS' CNN space (Liu et al., 2019) and the four DARTS subspaces S1-S4 (Zela et al., 2020a). Detailed experimental settings are in Appendix A.5.

## 5.1 COMPARISON WITH SOTA ON NAS-BENCH-201

Table 2 shows the average test error (%) of the competing approaches and our Zero-Cost-PT on the three datasets in NAS-Bench-201. Here we include the naive random search and original DARTS as baselines, and compare our approach with the recent zero-cost NAS algorithm NASWOT (Mellor et al., 2021), TE-NAS (Chen et al., 2021), as well as the perturbation-based NAS approaches DARTS-PT and DARTS-PT (fix $\alpha$) (Wang et al., 2021). As in all competing approaches, we perform search on CIFAR-10 and evalu-

Table 5: Comparison with the state-of-the-art differentiable NAS methods on the DARTS CNN search space (ImageNet).

| Method | Error [%] | | Params [M] | Cost [GPU-days] |
|---|---|---|---|---|
| | Top-1 | Top-5 | | |
| DARTS | 26.7 | 8.7 | 4.7 | 0.4 |
| SDARTS-RS | 25.6 | 8.2 | - | 0.4 |
| DARTS-PT | 25.5 | 8.0 | 4.6 | 0.8 |
| PC-DARTS | 25.1 | 7.8 | 5.3 | 0.1 |
| SGAS | **24.1** | 7.3 | 5.4 | 0.25 |
| TE-NAS | 26.2 | 8.3 | 6.3 | 0.05 |
| Zero-Cost-PT[1](best) | 24.4 | 7.5 | 6.3 | **0.018** |
| Zero-Cost-PT[1](4 seeds) | $24.6_{\pm 0.13}$ | $7.6_{\pm 0.09}$ | 6.3 | **0.018** |

[1] We use the same training pipeline from DARTS (Liu et al., 2019).

ate the final model on all three datasets. We see that on all datasets our Zero-Cost-PT (with `nwot`) consistently offers superior performance, especially on CIFAR-100 and ImageNet-16. On the other hand, the best existing perturbation-based algorithm, DARTS-PT (fix $\alpha$), fails on those two datasets, producing suboptimal results with small improvements compared to random search, suggesting that architectures discovered by DARTS-PT might not transfer well to other datasets. TE-NAS is second best on CIFAR but as we show in § 5.2, performance deteriorates on larger datasets like ImageNet.

## 5.2 DARTS CNN SEARCH SPACE

We now move to the much larger DARTS CNN search space. We use the same settings as in DARTS-PT (Wang et al., 2021), but instead of pre-training the supernet and fine-tuning it after each perturbation, we take an untrained supernet and directly perform Zero-Cost-PT algorithm as in Section 4.1. As in previous experiments, we run with N=10 architecture proposal iterations and V=100 validation iteration, using the same random seeds as in DARTS-PT. We then train the selected four architectures under different initializations (seeds 0-3) for 600 epochs, and report both the best and average test errors on both CIFAR-10 and ImageNet. Experimental details, discovered architectures and additional baselines can be found the in Appendix A.5, A.10 and A.6.

**Results on CIFAR-10.** As shown in Table 4 the proposed Zero-Cost-PT approaches can achieve much better average test error then the original DARTS and comparable to its newer variants SDARTS-RS (Chen & Hsieh, 2020) and SGAS (Li et al., 2020) at a much lower searching cost (especially when using `random` edge ordering). There is a significant search cost reduction compared to DARTS-PT. While DARTS-PT needs to perform retraining between iterations, Zero-Cost-PT only evaluates the score of the perturbed supernet with zero-cost proxies ($S_{\texttt{nwot}}$), requiring no more than a minibatch of data. Note that here the cost of Zero-Cost-PT reported in Table 4 is for N=10 architecture proposal iterations ( `random` edge discretization order), and thus a single proposal iteration only takes about a few minutes to run. The other variant Zero-Cost-PT$_{\texttt{global-op-iter}}$ offers better performance with lower variance compared to `random` but incurs slightly heavier computation.

**Results on ImageNet.** Table 5 shows the ImageNet classification accuracy for architectures searched on CIFAR-10. Our Zero-Cost-PT$_{\texttt{random}}$ algorithm is able to find architectures with a comparable accuracy much faster than previous work, further reinforcing its efficacy in this setting. While TE-NAS results on CIFAR-10 were very close to Zero-Cost-PT, a much larger difference is observed on ImageNet with an accuracy drop of 1.8 pp and a runtime that is ~2.5× slower than Zero-Cost-PT.

## 5.3 ROBUSTNESS ANALYSIS

It is well known that DARTS could generate trivial architectures with degenerative performance in certain cases. Zela et al. (2020a) have designed various special search spaces for DARTS to investigate its failure cases on them. As in DARTS-PT, we consider spaces S1-S4 to validate the robustness of Zero-Cost-PT in a controlled environment (detailed specifications can be found in Appendix A.4). As shown in Table 6, our approach consistently outperforms the original DARTS, the state-of-the-art DARTS-PT and DARTS-PT(fix $\alpha$) across S1 to S3 on both datasets CIFAR-10 and CIFAR-100, while on SVHN it offers competitive performance comparing the competing algorithms (best in S1, second best in space S2/S3 with .08/.02% gap). This confirms that our Zero-Cost-PT is robust in finding good performing architectures in spaces where DARTS typically fails, e.g. it has been shown (Wang et al.,

Table 6: Comparison in test error (%) with state-of-the-art perturbation-based NAS on DARTS spaces S1-S4.

| Space | DARTS[1] | DARTS-PT[1] | | Zero-Cost-PT[2] | |
|---|---|---|---|---|---|
| | Best | Best | Best (fix $\alpha$) | Avg. | Best |
| **CIFAR-10** | | | | | |
| S1 | 3.84 | 3.5 | 2.86 | $\mathbf{2.75}_{\pm 0.28}$ | **2.55** |
| S2 | 4.85 | 2.79 | 2.59 | $\mathbf{2.49}_{\pm 0.05}$ | **2.45** |
| S3 | 3.34 | 2.49 | 2.52 | $\mathbf{2.47}_{\pm 0.09}$ | **2.40** |
| S4 | 7.20 | 2.64 | **2.58** | $5.23_{\pm 0.76}$ | 4.69 |
| **CIFAR-100** | | | | | |
| S1 | 29.64 | 24.48 | 24.4 | $\mathbf{22.05}_{\pm 0.29}$ | **21.84** |
| S2 | 26.05 | 23.16 | 23.3 | $\mathbf{20.97}_{\pm 0.50}$ | **20.61** |
| S3 | 28.9 | 22.03 | 21.94 | $\mathbf{21.02}_{\pm 0.57}$ | **20.61** |
| S4 | 22.85 | 20.80 | **20.66** | $25.70_{\pm 0.01}$ | 25.69 |
| **SVHN** | | | | | |
| S1 | 4.58 | 2.62 | 2.39 | $\mathbf{2.37}_{\pm 0.06}$ | **2.33** |
| S2 | 3.53 | 2.53 | **2.32** | $2.40_{\pm 0.05}$ | 2.36 |
| S3 | 3.41 | 2.42 | 2.32 | $2.34_{\pm 0.05}$ | **2.30** |
| S4 | 3.05 | 2.42 | **2.39** | $2.83_{\pm 0.06}$ | 2.79 |

[1] Results taken from (Wang et al., 2021).
[2] Results obtained using random seeds 0 and 2.

2021) that in S2 DARTS tends to produce trivial architectures saturated with skip connections. On the other hand, we observe that Zero-Cost-PT doesn't perform well in search space S4, struggling with operation `noise`, which simply outputs a random Gaussian noise $\mathcal{N}(0,1)$ regardless of the input. This is expected as score $S(\mathcal{A}_{\setminus o})$ can be completely random if $o = \texttt{noise}$. However, since `noise` operation is not useful in NAS, we are satisfied with the robustness of Zero-Cost-PT on S1-S3.

## 6 CONCLUSION

In this paper, we formalized the implicit operation scoring proxies that are present within differentiable NAS algorithms to both analyze existing methods and propose new ones. We showed that lightweight operation scoring methods based on zero-cost proxies empirically outperform existing operation scoring functions such as DARTS Liu et al. (2019), DARTS-PT Wang et al. (2021) and TE-NAS (Chen et al., 2021). We also found that perturbation is more effective than discretization when scoring an operation, leading to our lightweight NAS algorithm, Zero-Cost-PT. Our approach outperforms the best available differentiable architecture search in terms of searching time and accuracy even in very large search spaces – something that was previously impossible with zero-cost proxies.

## 7 REPRODUCIBILITY STATEMENT

In our paper, we adhere to the NAS best practice checklist (see below) (Lindauer & Hutter, 2020) to ensure reproducibility. Our code (both NAS and training pipeline) is available in the supplementary material. We report all the details of our experimental setup, and describe all hyperparameters for both NAS and the final evaluation pipeline, as well as random seeds used in our main paper, appendix and code. We perform multiple runs of our experiments with the reported random seeds and provide average and standard deviation of the results. We report the wall-clock time for the search cost of our approach, and the training time for the discovered models are discussed in appendix. We also provide the detailed information on confounding factors, such as GPU hardware, versions of DL libraries, different runtimes of our experiments, in the appendix and the code (README in supplementary material).

When applicable we also compare to random search and other naive baselines (e.g. max-param models as detailed in appendix), and also use tabular benchmarks (NasBench-201) for in-depth evaluations and ablation study of our approach. For all existing NAS methods compared in this paper, we use exactly the same NAS benchmark as described in the literature and associated public repositories, including the same dataset (with the same training-test split), search space and code for training the architectures, and hyperparameters for that code. We use the same evaluation protocol for the methods being compared, as provided in their code repositories.

### 7.1 NAS BEST PRACTICE CHECKLIST

1. **Best Practices for Releasing Code**

    For all experiments you report:
    (a) Did you release code for the training pipeline used to evaluate the final architectures? [Yes]
    (b) Did you release code for the search space [Yes]
    (c) Did you release the hyperparameters used for the final evaluation pipeline, as well as random seeds? [Yes]
    (d) Did you release code for your NAS method? [Yes]
    (e) Did you release hyperparameters for your NAS method, as well as random seeds? [Yes]

2. **Best practices for comparing NAS methods**
    (a) For all NAS methods you compare, did you use exactly the same NAS benchmark, including the same dataset (with the same training-test split), search space and code for training the architectures and hyperparameters for that code? [Yes]
    (b) Did you control for confounding factors (different hardware, versions of DL libraries, different runtimes for the different methods)? [Yes]
    (c) Did you run ablation studies? [Yes]
    (d) Did you use the same evaluation protocol for the methods being compared? [Yes]
    (e) Did you compare performance over time? [N/A]
    (f) Did you compare to random search? [Yes]
    (g) Did you perform multiple runs of your experiments and report seeds? [Yes]
    (h) Did you use tabular or surrogate benchmarks for in-depth evaluations? [Yes]

3. **Best practices for reporting important details**
    (a) Did you report how you tuned hyperparameters, and what time and resources this required? [Yes]
    (b) Did you report the time for the entire end-to-end NAS method (rather than, e.g., only for the search phase)? [N/A]
    (c) Did you report all the details of your experimental setup? [Yes]

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

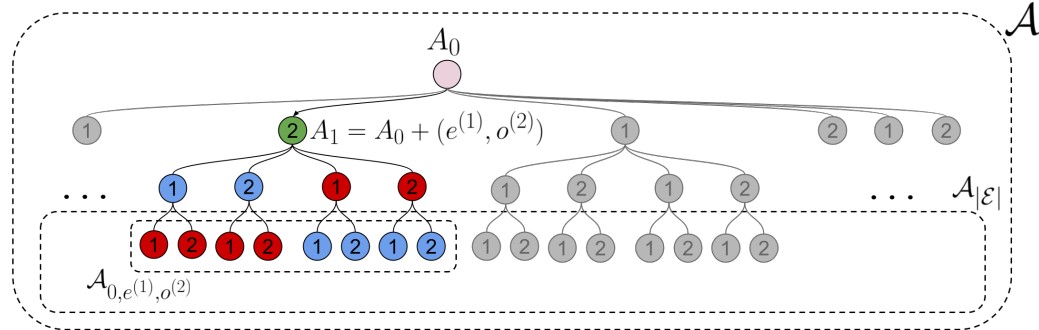

Figure A1: Visualisation of the searching tree related to a hypothetical supernetwork with 3 nodes (represented by different colors) and 2 candidate operations (represented by different numbers). The root node of the tree represents a full supernetwork. Each descendant node represents a non-unique network architecture obtained by discretizing an edge identified by the color of the node, with an operations encoded by a number in the node, given an architecture represented by a parent node. $\mathcal{A}$ is the set of all possible architectures, including the supernet $A_0$, partially-discretized networks like $A_1$ and fully-discretized networks. The set of all fully-discretized networks is additionally denoted as $\mathcal{A}_{|\mathcal{E}|}$, and the set of all fully-discretized networks that are achievable after operation $o^{(2)}$ is assigned to edge $e^{(1)}$, given $A_0$, is labeled as $\mathcal{A}_{0,e^{(1)},o^{(2)}}$. The arrow from $A_0$ to $A_1$ represents a possible first step during the discretization process which closes all grayed-out configurations. While performing NAS, at each tree level we need to decide which edge to follow; this is achieved by assigning scores to all/some of the edges that we are willing to consider in each iteration using different scoring functions which are the main focus of our analysis in Section 3.

# A  APPENDIX

## A.1  EXTRA DETSILS ABOUT NOTATION USED IN THE PAPER

All sets are denoted with stylised capital letters using latex's `matcal` font. Letters denoting elements of different sets use the same letters as the sets, e.g. $e \in \mathcal{E}$, $A \in \mathcal{A}$, etc. For any element, we use subscript for indexing iterations of the discretization process – e.g., $A_0$ is a network architecture at the beginning of iteration 0, $e_2$ is an edge that is being investigated in iteration 2, etc. To identify different elements in any other context we use superscript, e.g. $e^{(1)}$ might denote the first edge in a supernet, which might be different from $e_1$ which denotes the edge that is first going to be dicretized following a relevant discretization order.

To better understand meaning of each used symbol, consider a hypothetical supernet with 3 edges – $e^{(1)}, e^{(2)}, e^{(3)}$ – repented by different colors (green, blue and red, respectively) and 2 candidate operations – $o^{(1)}, o^{(2)}$ – represented by different numbers (1 and 2). Figure A1 visualises the entire space related to the decision process that is happening in order to perform NAS in this setting, including the first discretization step $A_1 = A_0 + (e_0, o_0)$, where $e_0 = e^{(1)}, o_0 = o^{(2)}$, and the related set of all achievable fully-discretized models $\mathcal{A}_{0,e_0,o_0}$ – a concept central to our definition of the optimal scoring function.

## A.2  DETAILED ZERO-COST-PT ALGORITHM

Algorithm 1 presents the proposed Zero-Cost-PT algorithm introduced in Section 4.1. It has two stages: *searching* and *validation*, where we first iteratively discretize the supernet $A_0$ based on zero-cost-based perturbation function $f_{\text{zc-pt}}$ (line 1 - 16), and then in the second stage we use the a zero-cost metric (the same which is used in $f_{\text{zc-pt}}$) to score the candidate architectures (line 17 - 20), and select the one with the highest end-to-end score. In particular for DARTS CNN search space, our Zero-Cost-PT algorithm has an additional topology selection step (line 8 - 14), where for each node in architecture we only retain the top two incoming edges based on the zc-pt score – this is similar to the vanilla DARTS algorithm (Liu et al., 2019). For NAS-Bench-201 space our algorithm skips this topology selection step.

---

**Algorithm 1:** Zero-Cost Perturbation-based Architecture Search (Zero-Cost-PT)

---

**Input :** An *untrained* supernetwork $A_0$ with set of edges $\mathcal{E}$ and set of nodes $\mathcal{N}$, # of architecture proposal
iterations N, # of validation iterations V

**Result:** A selected architecture $A^*_{|\mathcal{E}|}$

```
// Stage 1:  propose architecture candidates
```
1   $\mathcal{C} = \varnothing$
2   **for** $i = 1 : \text{N}$ **do**
3     **for** $t = 1 : |\mathcal{E}|$ **do**
4       Select next edge $e_t$ using the chosen discretization ordering
5       $o_t = \pi_{\text{zc-pt}}(A_{t-1}, e_t)$
6       $A_t = A_{t-1} + (e_t, o_t)$
7     **end**
8     **while** $|\mathcal{N}| > 0$ **do** `// prune the edges of the obtained architecture` $A_{|\mathcal{E}|}$
9       Randomly select a node $n \in \mathcal{N}$
10      **forall** Input edge $e$ to node $n$ **do**
11        Evaluate the zc-pt score of the architecture $A_{|\mathcal{E}|}$ when $e$ is removed
12      **end**
13      Retain only edges $e_n^{(1)*}$, $e_n^{(2)*}$ with the 1st and 2nd best zc-pt score, and remove $n$ from $\mathcal{N}$
14     **end**
15     Add $A_{|\mathcal{E}|}$ to the set of candidate architectures $\mathcal{C}$
16   **end**
```
// Stage 2:  validate the architecture candidates
```
17   **for** $j = 1 : \text{V}$ **do**
18     Calculate $S^{(j)}(A)$ for each $A \in \mathcal{C}$ using a random mini-batch data;
19   **end**
20   Select the best architecture $A^*_{|\mathcal{E}|} = \arg\max_{A \in \mathcal{C}} \sum_{j=1:\text{V}} S^{(j)}(A)$;

---

### A.3   MORE ON OPERATION SCORING

This section provides more experimental details and examples of our analysis of the operation scoring functions introduced in Section 3.

#### A.3.1   DETAILED SCORING METHODOLOGY

As discussed in Section 3.2.1, our analysis on the on the initial operation scoring aims to investigate how well an operation scoring method can perform when making the first discretization decision, with respect to the perfect search (the best-acc approach). Here "the first discretization decision" is made at the first iteration (iteration 0) of a progressive operation selection algorithm, and in our experiment we compute the score for all operations on an edge and later average across all edges to account for random selection of the first edge. Concretely, we compute the score per operation across all edges, then compute the Spearman rank correlation for operations on each edge. After that, we average the rank correlation coefficient over all edges.

Consider the example shown in Figure A2. Suppose we have a supernet with just two edges as in Figure A2. In this case, an operation scoring function should pick just one operation per edge. For a given operation scoring function, we compute the scores for each operation on each edge. Then for each edge, we compute the rank correlation of the oracle scores (best-acc or avg-acc) against the scores from the other operation scoring function (e.g. zc-pt or darts-pt). We then average their correlation coefficient across all edges in a supernet to get an average correlation for each operation scoring function. The resulting average rank correlation is indicative of how well a given operation scoring function (starting with a random edge) would do when making the first discretization decision, relative to the oracle search.

#### A.3.2   EXPERIMENTAL DETAILS

Here, we provide some additional experimental details for the data presented in Section 3. The following list describes how we compute each operation score.

- **best-acc**: To get the score for an operation $o$ on a specific edge $e$, we find the maximum test accuracy of all NAS-Bench-201 architectures with $(o, e)$.

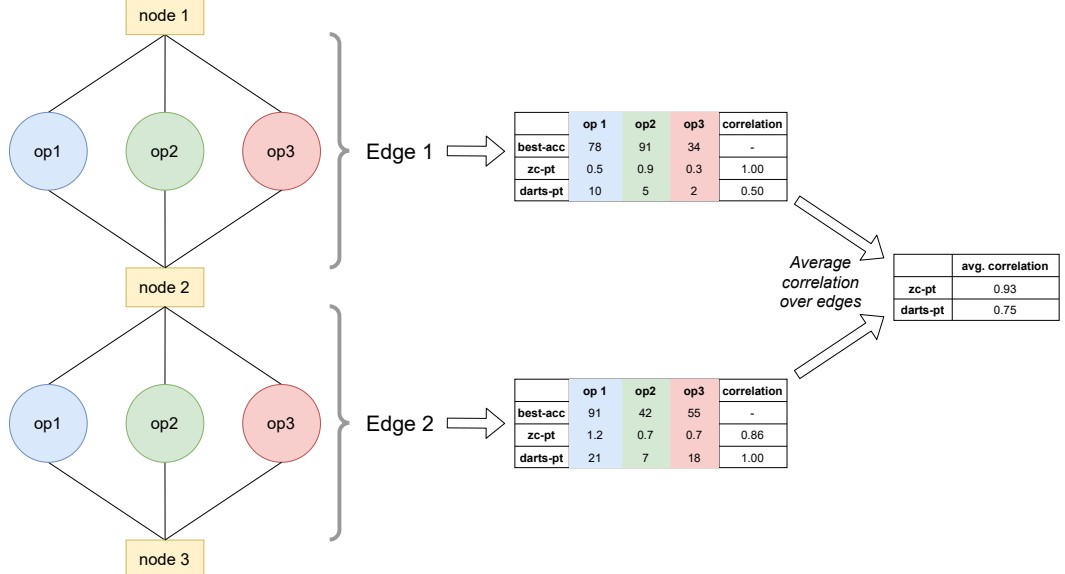

Figure A2: An example showing how correlations of different proxies are computed in our analysis on initial operation scoring in Section 3.2.1

- **avg-acc**: Same as best-acc but we *average* all NAS-Bench-201 architecture test accuracies instead of finding the maximum.

- **disc-acc**: We discretize one edge $e$ by selecting an operation $o$, then we train for 5 epochs[3] and record the supernet accuracy – this is used as the score for $(o, e)$.

- **darts-pt**: We perturb one edge with one operation $A - (e, o)$ and record the validation accuracy. For perturbation-based scoring functions, we multiply the score by $-1$ before computing correlations.

- **disc-zc**: We discretize one edge $e$ by selecting an operation $o$ and then compute the zero-cost metric.

- **zc-pt**: We perturb one edge with one operation $A - (e, o)$ and compute the zero-cost metric. For perturbation-based scoring functions, we multiply the score by $-1$ before computing correlations.

- **darts**: We record the value of the architecture parameters $\alpha$ after 60 epochs of training the supernet.

- **tenas**: We perturb one edge with one operation $A - (e, o)$ and compute the $\kappa_{\mathcal{N}}$ and the $\hat{R}_{\mathcal{N}}$ (number of linear regions). Then we rank $\kappa_{\mathcal{N}}$ ascendingly and descendingly rank the $\hat{R}_{\mathcal{N}}$. At last, we add those two ranks together to get the final ranks of the operations. We multiply the score by $-1$ before computing correlations.

### A.3.3 Detailed Operation Scores

Table A8 (at the end of this Appendix) shows all operation scores at iteration 0. This data was used to compute Spearman-$\rho$ in Figure 2a. Note that we compute Spearman-$\rho$ per edge and average over all edges – this summarizes how well each score tracks our "oracle" best-acc score.

### A.4 Description of DARTS subspaces (S1-S4)

RobustDARTS introduced four different DARTS subspaces to evaluate the robustness of the original DARTS algorithm (Zela et al., 2020a). In our work, we validate the robustness of Zero-Cost-PT against some of the more recent algorithms using the same subspaces originally proposed in the RobustDARTS paper (Section 5.3). The search spaces are defined as follows:

---

[3]DARTS-PT defines discretization accuracy as the accuracy after *convergence*. We elected to only train for 5 epochs to make our experiments feasible but we are now investigating whether longer training will affect our results.

- in S1 each edge of a supernet consists only of the two candidate operations having the highest magnitude of $\alpha$ in the vanilla DARTS (these operations can be different for different edges);

- S2 only considers two operations: `skip_connect` and `sep_conv_3x3`;

- similarly, S3 consists of three operations: `none`, `skip_connect` and `sep_conv_3x3`;

- finally, S4 again considers just two operation: `noise` and `sep_conv_3x3`, where `noise` operation generates random Gaussian noise $\mathcal{N}(0,1)$ in every forwards pass that is independent from the input.

## A.5  EXPERIMENTAL DETAILS

All searches were run multiple times with different searching seeds (usually 0, 1, 2 and 3). Additionally, each found architecture was trained multiple times using different training seeds – for DARTS the same set of seeds was used for training and searching, for NAS-Bench-201 (NB201) training seeds were taken from the dataset (777, 888 and 999, based on their availability in the dataset). Therefore, for each experiment we got a total of `searching_seeds × training_seeds` accuracy values. Whenever average performance is reported, it is averaged across all obtained results. Similarly, best values were selected by taking the best single result from the searching and training seeds.

### A.5.1  EXPERIMENTAL DETAILS – NAS-BENCH-201

Searching was performed using 4 different seeds (0, 1, 2, and 3) to initialise a supernet. Whenever we had to perform training of a supernet during the searching phase (Section 3), we used the same hyperparameters as the original DARTS-PT code used. When searching using our Zero-Cost-PT we used batch size of 256, N=10, V=100 and S=`nwot`, unless mentioned otherwise (e.g., during ablation studies). Inputs for calculating zero-cost scores came from the training dataloader(s), as defined for relevant datasets in the original DARTS-PT code (including augmentation). For zero-cost proxies that require a loss function, standard cross-entropy was used. For any searching method, after an algorithm had identified a final subnetwork, we extracted the final architecture and queried the NB201 dataset to obtain test accuracy – one value for each training seed available in the dataset.

All experiments concerning operation scoring (Sections 3 and A.3) used averaged accuracy of models from NB201 for simplicity.

We did not search for architectures targeting CIFAR-100 or ImageNet-16 directly – whenever we report results for these datasets, we used the same architecture found using CIFAR-10.

### A.5.2  EXPERIMENTAL DETAILS – DARTS

DARTS experiments follow a similar methodology to NB201. Each algorithm was run with 4 different initialisation seeds for a supernet (0, 1, 2 and 3). When running Zero-Cost-PT, we used the following hyperparameters: batch size of 64, N=10, V=100 and S=`nwot`. Inputs and loss function for zero-cost metrics were defined analogically to NB201. We did not run any baseline method on the DARTS search space (all results were taken from the literature), so we did not have to perform any train a supernet. After an algorithm had identified a final subnetwork, we then trained it 4 times using different initialisation seeds again (0, 1, 2 and 3). When training subnetworks, we used a setting aligned with the previous work (Liu et al., 2019; Wang et al., 2021).

Unlike NB201, whenever different datasets were considered, (Section 5.3) architectures were searched on each relevant dataset directly.

For CIFAR-10 experiments, we trained models using a heavy configuration with `init_channels` = 36 and `layers` = 20. Models found on CIFAR-100 and SVHN were trained using a mobile setting with `init_channels` = 16 and `layers` = 8. Both choices follow the previous work (Zela et al., 2020a; Wang et al., 2021).

## A.6  MORE BASELINES

### A.6.1  MAXIMUM-PARAM BASELINE

As shown in Table 4 and Table 5 in Section 5.2, our approach tends to select architectures with comparable or slightly more parameters than the state-of-the-art. However, this does not mean any larger models would lead to superior performance, i.e. simply maximizing FLOPs/Params is not an

appropriate searching methodology in general. Below are the results of training 4 random models with separable convolution 5x5 (the most expensive operation in the DARTS search space) selected everywhere and random connections between layers. The evaluation methodology follows all other experiments, i.e. we have 4 searched models trained 4 times with different training seeds, and we report the average and minimum error.

Overall, the test error (%) of this method is $2.93_{\pm 0.23}$ (avg.) and 2.78 (min), vs. ours $2.64_{\pm 0.16}$ (avg.) and 2.43 (min). This confirms that simply selecting models with maximu FLOPs/Params is not an appropriate searching methodology in general, as evidenced by the above results on the DARTS search spaces. We also show that the results translate to ImageNet in later sections. On the other hand, on simple search spaces like NAS-Bench-201 (NB201), this maximum-param baseline may perform relatively better, e.g. it can find the model with test accuracy of

Table A1: Randomly selected architectures with only operation `sep_conv_5x5` on DARTS CNN space.

| S. seed [1] | Test Error(%) | | | | | |
| | Training seed [2] | | | | Avg. | Best |
| | 0 | 1 | 2 | 3 | | |
|---|---|---|---|---|---|---|
| 0 | 3.07 | 2.93 | 2.89 | 2.85 | $2.94_{\pm 0.30}$ | 2.85 |
| 1 | 2.92 | 2.93 | 3.17 | 3.05 | $3.02_{\pm 0.20}$ | 2.92 |
| 2 | 2.98 | 2.97 | 2.93 | 2.90 | $2.95_{\pm 0.06}$ | 2.95 |
| 3 | 2.87 | 2.83 | 3.02 | 2.78 | $2.88_{\pm 0.18}$ | 2.78 |

[1] Random seeds for searching the architectures.
[2] Random seeds for training the selected architectures.

93.76% (165th position in the ranking), but our method can still do better, discovering the 33rd best model in the search space (test accuracy 94.03%).

In summary, on both NB201 and DARTS space, the proposed combination of the perturbation paradigm with zero-shot proxies does better than the naive usage of the proxies presented in (Abdelfattah et al., 2021; Mellor et al., 2021), as shown in Tables 2, 4, 5 and A2. Note that we are better than the plain perturbation-based baseline (Wang et al., 2021) and recent zero-cost NAS (Chen et al., 2021) on NB201 (Table 2), and comparable/better on DARTS and derived subspaces (Table 4 and 5), while being much cheaper to run.

The structures of models are provided as Genotype objects below for reproducibility.

```
random_max_0 = Genotype(normal=[["sep_conv_5x5", 0], ["sep_conv_5x5", 1],
    ["sep_conv_5x5", 0], ["sep_conv_5x5", 1], ["sep_conv_5x5", 0], ["
    sep_conv_5x5", 3], ["sep_conv_5x5", 1], ["sep_conv_5x5", 4]],
    normal_concat=range(2, 6), reduce=[["sep_conv_5x5", 0], ["
    sep_conv_5x5", 1], ["sep_conv_5x5", 0], ["sep_conv_5x5", 1], ["
    sep_conv_5x5", 0], ["sep_conv_5x5", 1], ["sep_conv_5x5", 0], ["
    sep_conv_5x5", 1]], reduce_concat=range(2, 6))
random_max_1 = Genotype(normal=[["sep_conv_5x5", 0], ["sep_conv_5x5", 1],
    ["sep_conv_5x5", 0], ["sep_conv_5x5", 2], ["sep_conv_5x5", 0], ["
    sep_conv_5x5", 2], ["sep_conv_5x5", 1], ["sep_conv_5x5", 4]],
    normal_concat=range(2, 6), reduce=[["sep_conv_5x5", 0], ["
    sep_conv_5x5", 1], ["sep_conv_5x5", 0], ["sep_conv_5x5", 1], ["
    sep_conv_5x5", 0], ["sep_conv_5x5", 1], ["sep_conv_5x5", 0], ["
    sep_conv_5x5", 1]], reduce_concat=range(2, 6))
random_max_2 = Genotype(normal=[["sep_conv_5x5", 0], ["sep_conv_5x5", 1],
    ["sep_conv_5x5", 0], ["sep_conv_5x5", 2], ["sep_conv_5x5", 0], ["
    sep_conv_5x5", 3], ["sep_conv_5x5", 1], ["sep_conv_5x5", 4]],
    normal_concat=range(2, 6), reduce=[["sep_conv_5x5", 0], ["
    sep_conv_5x5", 1], ["sep_conv_5x5", 0], ["sep_conv_5x5", 1], ["
    sep_conv_5x5", 0], ["sep_conv_5x5", 1], ["sep_conv_5x5", 0], ["
    sep_conv_5x5", 1]], reduce_concat=range(2, 6))
random_max_3 = Genotype(normal=[["sep_conv_5x5", 0], ["sep_conv_5x5", 1],
    ["sep_conv_5x5", 0], ["sep_conv_5x5", 1], ["sep_conv_5x5", 0], ["
    sep_conv_5x5", 2], ["sep_conv_5x5", 1], ["sep_conv_5x5", 4]],
    normal_concat=range(2, 6), reduce=[["sep_conv_5x5", 0], ["
    sep_conv_5x5", 1], ["sep_conv_5x5", 0], ["sep_conv_5x5", 1], ["
    sep_conv_5x5", 0], ["sep_conv_5x5", 1], ["sep_conv_5x5", 0], ["
    sep_conv_5x5", 1]], reduce_concat=range(2, 6))
```

### A.6.2    RANDOM-SAMPLING BASELINE

In Section 5.1, we compared our method to sampling-based zero-cost NAS in Table 2 (see NAS-WOT lines). Our results are empirically better on all three datasets. Additionally, our method computes the operation score per edge in a supernet, whereas the sampling-based approach computes the end-to-end network score. The relationship between the number of subnetworks and the number of operations is exponential. Therefore, we anticipate having to sample exponentially many networks in sample-based NASWOT (Mellor et al., 2021) compared to our proposed Zero-Cost-PT.

Table A2: Comparison with randomly sampled networks in DARTS CNN space (CIFAR-10).

| Sample Size | Test Error(%) | | Cost (GPU-Days) |
|---|---|---|---|
| | Avg. | Best | |
| 2500 | $2.99_{\pm 0.22}$ | 2.66 | 0.018 |
| 20000 | $2.73_{\pm 0.09}$ | 2.58 | 0.083 |
| 50000 | $2.70_{\pm 0.09}$ | 2.52 | 0.208 |
| Zero-cost-PT | $2.64_{\pm 0.16}$ | 2.43 | 0.018 |

In order to extend the comparison between zero-cost NAS (NASWOT) and our zero-cost PT to the DARTS CNN search space, we have conducted further experiments, in which we allow NASWOT to sample and score random models from the DARTS search space for for a specified amount of samples (2500, 20000, 50000, corresponding to roughly 25min, 2h, and 5h on a single 2080Ti GPU) and the best model, according to the `nwot` metric, is selected as the result of a search. For each of the three sampling budgets, we run the entire process 4 times using different searching seeds (0-3), thus resulting in $3 \times 4 = 12$ final architectures (9 unique ones are selected). Each of the final architectures was then trained 4 times using different training seed (0-3). The results are presented in details in Table A2.

As expected, both average and best performance of the sample-based zero-cost search increases with more samples. However, the results are visibly behind our proposed method, even for the most expensive searches (5 hours). In addition, we see that increasing searching budget from 2 hours to 5 hours does not result in a proportional gains in accuracy, compared to 25min, suggesting diminishing returns. We hypothesize it is related to the mentioned fact that the number of architectures grows exponentially, so we'd need to sample significantly more networks before the probability of hitting a good one increases noticeably. In fact, we find that when increasing searching budget from 20000 to 50000 samples in our experiments, the baseline only get better results in one out of 4 cases (searching seeds), and in one it actually made the results worse. This further suggests diminishing returns. Finally, for a similar time budget to ours (25 min), the average performance of the baseline is actually much closer to the random search ($3.29_{\pm 0.15}$) (Liu et al., 2019) than to our method, with significant variance.

### A.6.3 ZERO-COST-DISC BASELINE

In Section. 3, we propose two zero-cost operation scoring function $\pi_{\text{disc-zc}}$ and $\pi_{\text{zc-pt}}$, and study their correlation with an oracle metric best-acc on NAS-Bench-201 (Dong & Yang, 2020). We find that discretization is generally a weaker scoring paradigm than perturbation, as shown by their correlations with respect to the oracle score best-acc. To further evaluate the end-to-end NAS perfor-

Table A3: Comparison in test error (%) between zero-cost perturbation-based and discretization-based NAS on NAS-Bench-201.

| Method[1] | CIFAR-10 | CIFAR-100 | ImageNet-16 |
|---|---|---|---|
| **Zero-Cost-DISC** | $6.22_{\pm 0.84}$ | $28.18_{\pm 2.01}$ | $55.14_{\pm 1.77}$ |
| **Zero-Cost-PT** | $\mathbf{5.97_{\pm 0.17}}$ | $\mathbf{27.47_{\pm 0.28}}$ | $\mathbf{53.82_{\pm 0.77}}$ |

[1] We use the same hyperparameter settings as reported in the main paper: N=10, V=100, `nwot` zero-cost metric and `random` edge discretization order.

mance of discretization vs. perturbation with zero-cost metrics, we consider a baseline named Zero-Cost-DISC, which discretizes the supernet based on $\pi_{\text{disc-zc}}$ instead of $\pi_{\text{zc-pt}}$. Details on how disc-zc computes the operation scores can be found in Appendix A.3.2. We compare the performance of Zero-Cost-DISC and our proposed Zero-Cost-PT on NAS-Bench-201 (Dong & Yang, 2020), as shown in Table A3. We see that discretization (Zero-Cost-DISC) results in inferior performance compared to the proposed perturbation-based approach (Zero-Cost-PT) on all datasets, confirming our previous analysis on their correlations with the oracle metric.

### A.7 ADDITIONAL ABLATION STUDY ON DARTS CNN SPACE

We conducted ablation study of our Zero-Cost-PT algorithm on NAS-Bench-201 (Dong & Yang, 2020) in Section. 4.2, aiming to decide the best possible configuration of the main hyperparameters of our algorithm: architecture proposal iterations N, validation iterations V, ordering of edges to follow

when discretizing, and the zero-cost metric to use. In the following we present additional ablations on the much larger DARTS CNN space, in particular to study the impact of different architecture proposal iterations $N$ and validation iterations $V$ when Zero-Cost-PT uses `random` as the search order and `nwot` metric.

As detailed in Algorithm 1, Zero-Cost-PT$_{random}$ firstly proposes $N$ candidate architectures using the proposed zero-cost perturbation paradigm, which are then evaluated in a lightweight manner during the validation phase to come up with a single outcome of a search. It extends the existing zero-cost NAS approach such as NASWOT (Mellor et al., 2021) by including more sophisticated architecture selection phase based on the combination of zero-cost metrics and perturbations.

We first consider an extreme case, setting architecture proposal iteration $N=1$, where Zero-Cost-PT only proposes one architecture candidate (with `random` edge discritization order), and with no validation stage performed. The detailed results are shown in Table A4. As can be seen, the average performance is affected quite significantly. However, the best model still happens to be on-par with our main results. This suggests what has already been mentioned in our main paper, that searching phase alone tends to "find" many different architectures depending on, broadly speaking, random seed, and this randomness is especially visible in the case of random edge ordering (Figure 3a). While the high variance might seem undesired at first, we empirically observe that the higher exploration resulting from it is beneficial for finding some very good models, e.g., `global-op-iter` discretization order tends to be less sensitive to random seed as it takes away one degree of randomness (edge order), producing more stable results on average, but at the same time limiting its ability to maximize performance of the best model found (Table 4). In order to maximize performance of our method, we balance exploration (higher $N$ + random edge order) and exploitation (higher $V$) in the searching and validation phase respectively.

Table A4: Detailed performance of Zero-Cost-PT$_{random}$ with $N=1$, $V=0$, `nwot` metric on DARTS CNN space.

| S. seed [1] | Test Error (%) | | | | | |
| --- | --- | --- | --- | --- | --- | --- |
| | Training seed [2] | | | | Avg. | Best |
| | 0 | 1 | 2 | 3 | | |
| 0 | 2.72 | 2.55 | 2.83 | 2.71 | | |
| 1 | 3.25 | 3.26 | 3.28 | 3.20 | $2.81_{\pm 0.29}$ | 2.43 |
| 2 | 2.59 | 2.84 | 2.59 | 2.79 | | |
| 3 | 2.43 | 2.77 | 2.52 | 2.66 | | |

[1] Random seeds for searching the architectures.
[2] Random seeds for training the selected architectures.

Table A5: Detailed performance of Zero-Cost-PT$_{random}$ with $N=10$, $V=\{1, 10, 100\}$, `nwot` metric on DARTS CNN space.

| V | S. seed [1] | Test Error (%) | | | | | |
| --- | --- | --- | --- | --- | --- | --- | --- |
| | | Training seed [2] | | | | Avg. | Best |
| | | 0 | 1 | 2 | 3 | | |
| 1 | 0 | 3.08 | 3.16 | 3.06 | 2.96 | | |
| | 1 | 2.74 | 2.91 | 2.92 | 2.92 | $2.93_{\pm 0.14}$ | 2.65 |
| | 2 | 2.96 | 3.10 | 3.06 | 2.90 | | |
| | 3 | 2.86 | 2.85 | 2.85 | 2.65 | | |
| 10 | 0 | 3.08 | 3.16 | 3.06 | 2.96 | | |
| | 1 | 2.74 | 2.91 | 2.92 | 2.87 | $2.88_{\pm 0.14}$ | 2.65 |
| | 2 | 2.77 | 2.71 | 2.65 | 2.76 | | |
| | 3 | 2.83 | 3.00 | 2.82 | 2.87 | | |
| 100 | 0 | 2.86 | 2.97 | 2.77 | 2.82 | | |
| | 1 | 2.51 | 2.47 | 2.56 | 2.43 | $2.64_{\pm 0.16}$ | 2.43 |
| | 2 | 2.74 | 2.73 | 2.54 | 2.62 | | |
| | 3 | 2.43 | 2.77 | 2.52 | 2.64 | | |

[1] Random seeds for searching the architectures.
[2] Random seeds for training the selected architectures.

Admittedly, the interplay between those two phases is crucial for our method. To further showcase how the validation phase complements the searching phase, we run additional ablations on the DARTS CNN space with $N=10$ and $V=\{1,10,100\}$, the results are shown in Table A5. The results are consistent with what is shown in the main paper: higher $V$ produces better results on average but does not affect the best case that much (the best model is still upper-bounded by what was found with $N=1$).

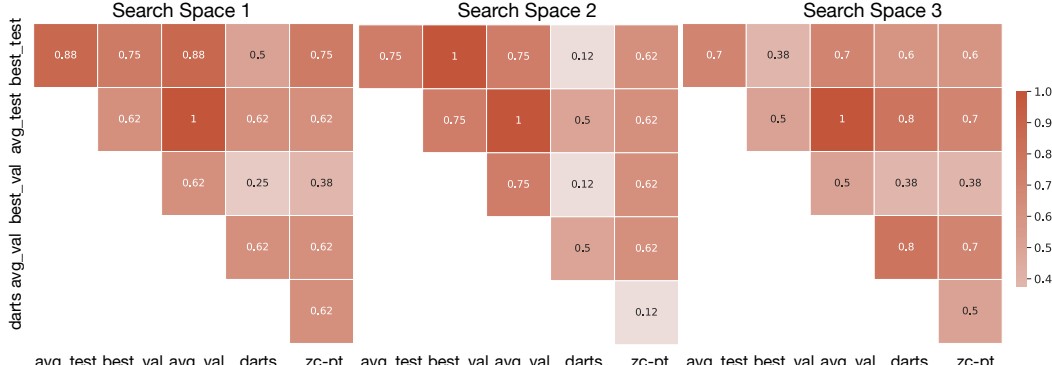

Figure A3: Spearman's rank correlation coefficient of different operation scoring metrics with each other at the first iteration of NAS evaluated on NAS-Bench-1Shot1 benchmark.

## A.8 ADDITIONAL EXPERIMENTS ON NAS-BENCH-1SHOT1

In addition to experiments on NAS-Bench-201 (Dong & Yang, 2020), DARTS CNN space (Liu et al., 2019) and DARTS subspaces S1-S4 (Wang et al., 2021), we perform additional experiments on NAS-Bench-1shot1 (Zela et al., 2020b) to further evaluate the performance of our Zero-Cost-PT algorithm. We first extend our correlation analysis from Section 3 (in particular, the initial operation scoring as in Section 3.2.1). We compare to DARTS as it is already available in the NB1shot1 codebase [4]. As in Figure A3, the results show that DARTS is surprisingly well-correlated to both best and avg accuracy, in some cases even better than our proposed ZC-PT. We believe that this is because the search space does not contain skip connections and overall is rather unusual compared to others used with differentiable NAS, so it is possible that it constitutes an edge-case where DARTS performs relatively well.

Table A6 shows the NAS results. We can see that DARTS retains its high performance. On the other hand, our ZC-PT achieves worse average performance, with noticeable variance (especially on the largest Space 3), but is also able to find better models in the best-case scenario. We would like to notice that unlike DARTS or NAS-Bench-201, NAS-Bench-1shot1 supernet contains architectural parameters associated with entire connections between cells ($\alpha$ and $\gamma$ in the paper (Zela et al., 2020b)), additionally to the standard ones associated with candidate operations in a single layer ($\beta$ in the paper). What it means for our method is that for those parameters we no longer perturb a single edge of a supernet but rather the entire path, making the setting for our algorithm noticeably different.

Table A6: Performance of DARTS (Liu et al., 2019) and our Zero-Cost-PT on NAS-Bench-1Shot1 (Zela et al., 2020b).

| Method | Space | Test Error (%) | |
| --- | --- | --- | --- |
| | | Avg. | Best |
| DARTS | 1 | $6.67_{\pm 0.08}$ | 6.44 |
| Zero-Cost-PT | | $6.76_{\pm 1.05}$ | 5.45 |
| DARTS | 2 | $6.68_{\pm 0.36}$ | 6.24 |
| Zero-Cost-PT | | $6.84_{\pm 0.40}$ | 6.14 |
| DARTS | 3 | $6.64_{\pm 0.12}$ | 6.5 |
| Zero-Cost-PT | | $7.45_{\pm 0.94}$ | 6.03 |

Further extending our method from operation scoring to path scoring is a very relevant goal for future work.

## A.9 ADDITIONAL EXPERIMENTS ON MOBILENET-LIKE SEARCH SPACE

In addition to the DARTS-like search spaces (including NAS-Bench-201, DARTS CNN Space, DARTS subspaces S1 - S4) studied in our paper, for completeness we also conduct additional experiments on the MobileNet-like search space, to verify the robustness and generalization capability of the proposed Zero-Cost-PT approach.

It is well known that most of the existing NAS algorithms designed for MobileNet-like space constrain the model #FLOPS/Params during the search process. However, our method is not designed for such constrained NAS context (like the original DARTS and DARTS-PT). To the best of our knowledge,

---

[4]https://github.com/automl/nasbench-1shot1

Table A7: Error and search cost of Zero-Cost-PT on MobileNet-like search space (ImageNet)

| Architecture | Top-1 Error (%) | Top-5 Error (%) | Params. (M) | Cost (GPU Days) |
|---|---|---|---|---|
| ProxylessNAS (GPU) | 24.9 | 7.5 | 7.1 | 8.3 |
| Zero-Cost-PT(seed 0) | 24.0 | 7.0 | 8.0 | 0.041 |
| Zero-Cost-PT(seed 1) | 23.6 | 6.8 | 8.0 | 0.041 |
| Zero-Cost-PT(seed 2) | 23.9 | 7.0 | 8.3 | 0.041 |

so far there is no clear solution on how one could constrain #FLOPS/Params of the resulting final architecture during the process of discritizing operations on edges of the supernet. Essentially, to do that, when selecting the operations on an edge of the supernet we need to consider both their scores and the potential contributions to the sum of FLOPS/Params of the final model, which is potentially a NP hard problem. Therefore, in our experiments on MobileNet-like search space, we do not enforce constraints on #FLOPS/Params during search, as it is less relevant to the proposed approach.

### A.9.1 EXPERIMENT SETTING

We adopt the same settings as in (Cai et al., 2019) and construct a supernet with 21 choice blocks, and each block has the following 7 alternative operations:

- 3×2 = 6 MobileNet blocks, with 3 different kernel sizes {3, 5, 7} and 2 expansion ratio {3, 6}).
- skip connection.

We follow the previous work (Cai et al., 2019) on this space and search directly on ImageNet dataset (Deng et al., 2009), with `input_size` = 224. We use 3 different random seeds (0-2) to perform architectures search, and train the discovered final models (3 models searched and trained in total). We use `batch-size` = 1024 and training was performed on 8 NVIDIA V100 GPUs for 300 epochs, with initial learning rate set to 0.5. We use colour-jitters, random horizontal flip and random crop for data augmentation, set `label_smooth` = 0.1. All other training setting is identical to ProxylessNAS (Cai et al., 2019).

### A.9.2 RESULTS

Table A7 shows the performance (error %) of the architectures discovered by the proposed Zero-Cost-PT algorithm on ImageNet. We see that comparing to ProxylessNAS, our approach can find models with comparable or better accuracy, with much less search cost. This further confirms the robustness and transferability of the Zero-Cost-PT approach, on a different search space in addition to the DARTS-like spaces studied in our paper. Noticeably the models found by our approach have higher #Params comparing to ProxylessNAS, but as we discussed above, constraining #FLOPS/Params is less relevant in our context, and the aim of this experiment is to show our Zero-Cost-PT can also discover good performing architectures on MobileNet-like space with low search cost.

### A.10 DISCOVERED ARCHITECTURES

Figures A4 and A5 present cells found by our Zero-Cost-PT on the DARTS CNN search space (Section 5.2) when using `global-op-iter` and `random` discretization orders, respectively (see Section 4.2 for the definition of the two discretization orders). Figures A7 through A18 show cells discovered on the four DARTS subspaces and the three relevant datasets (Sections 5.3 and A.4). Figure A19 shows architectures found by our Zero-Cost-PT approach on MobileNet-like search space.

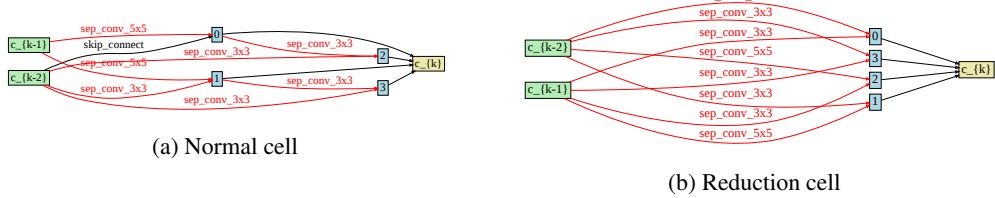

(a) Normal cell

(b) Reduction cell

Figure A4: Cells found by Zero-Cost-PT (`global-op-iter` discretization order) on the DARTS search space using CIFAR-10.

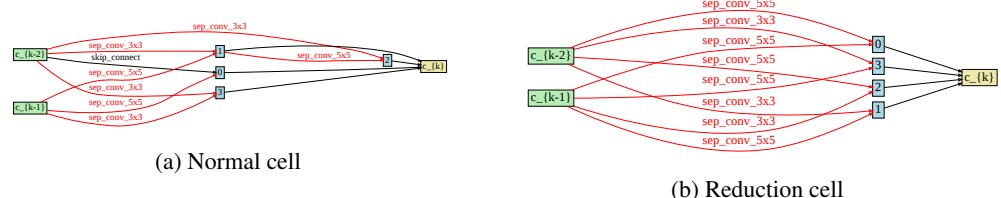

(a) Normal cell

(b) Reduction cell

Figure A5: Cells found by Zero-Cost-PT (`random` discretization order) on the DARTS search space using CIFAR-10.

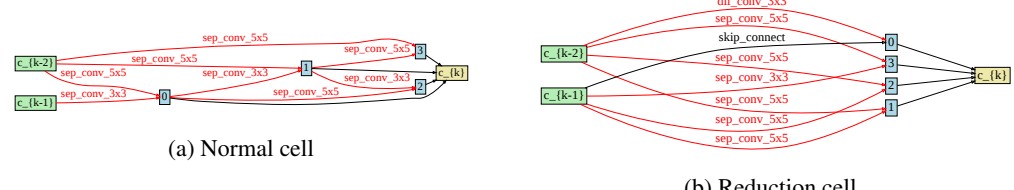

(a) Normal cell

(b) Reduction cell

Figure A6: Cells found by Zero-Cost-PT (`random` discretization order) on the DARTS search space using ImageNet.

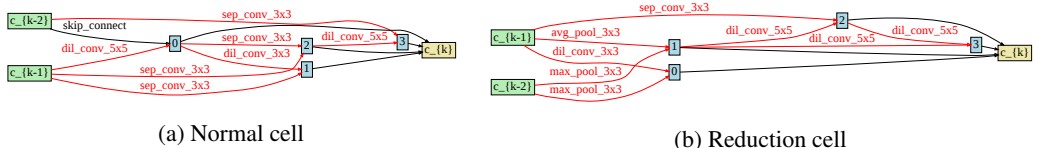

(a) Normal cell

(b) Reduction cell

Figure A7: Cells found by Zero-Cost-PT (`random` discretization order) on the DARTS-S1 space using CIFAR-10.

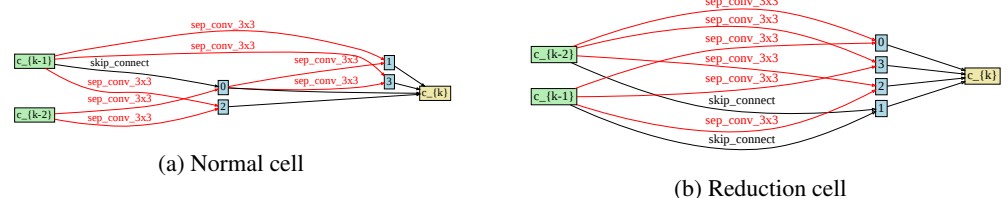

(a) Normal cell

(b) Reduction cell

Figure A8: Cells found by Zero-Cost-PT (`random` discretization order) on the DARTS-S2 space using CIFAR-10.

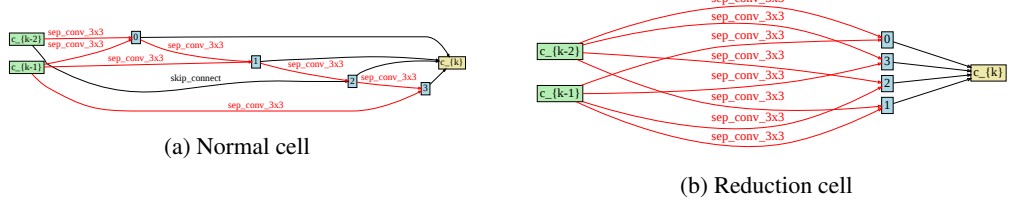

(a) Normal cell

(b) Reduction cell

Figure A9: Cells found by Zero-Cost-PT (`random` discretization order) on the DARTS-S3 space using CIFAR-10.

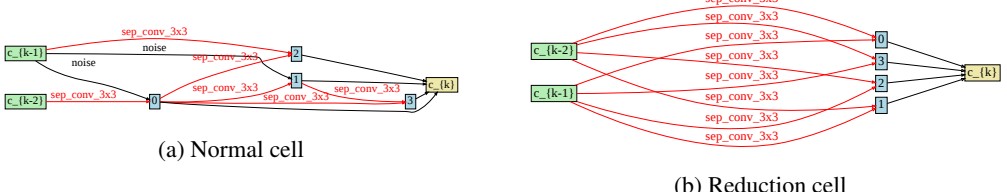

(a) Normal cell

(b) Reduction cell

Figure A10: Cells found by Zero-Cost-PT (`random` discretization order) on the DARTS-S4 space using CIFAR-10.

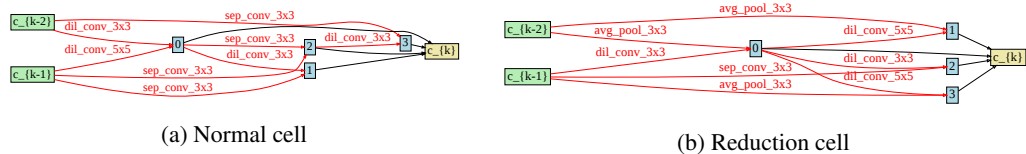

(a) Normal cell

(b) Reduction cell

Figure A11: Cells found by Zero-Cost-PT (`random` discretization order) on the DARTS-S1 space using CIFAR-100.

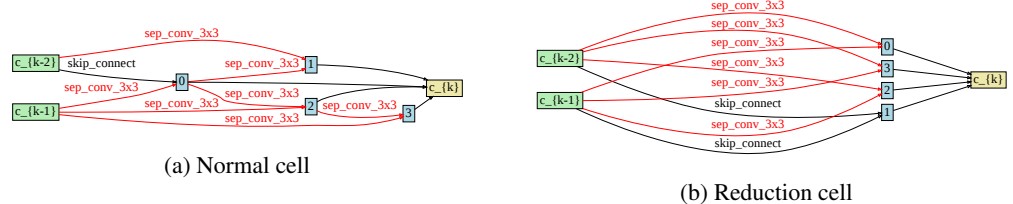

(a) Normal cell

(b) Reduction cell

Figure A12: Cells found by Zero-Cost-PT (`random` discretization order) on the DARTS-S2 space using CIFAR-100.

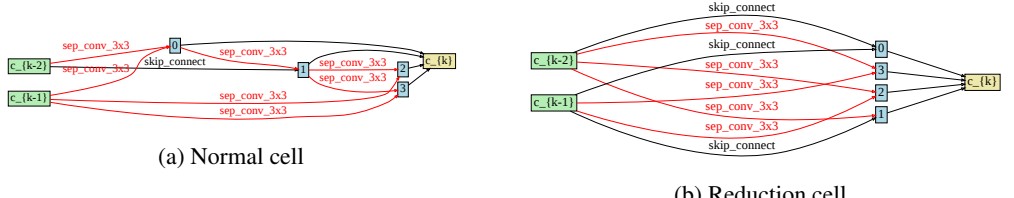

(a) Normal cell

(b) Reduction cell

Figure A13: Cells found by Zero-Cost-PT (`random` discretization order) on the DARTS-S3 space using CIFAR-100.

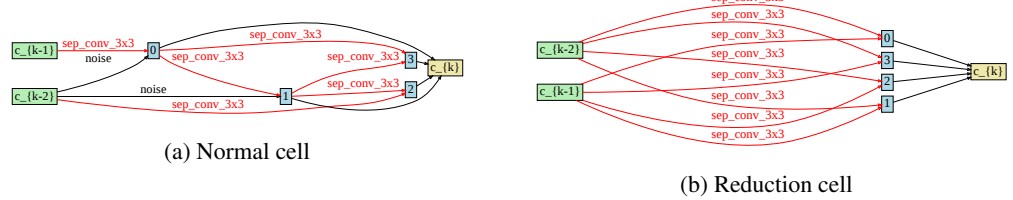

(a) Normal cell

(b) Reduction cell

Figure A14: Cells found by Zero-Cost-PT (`random` discretization order) on the DARTS-S4 space using CIFAR-100.

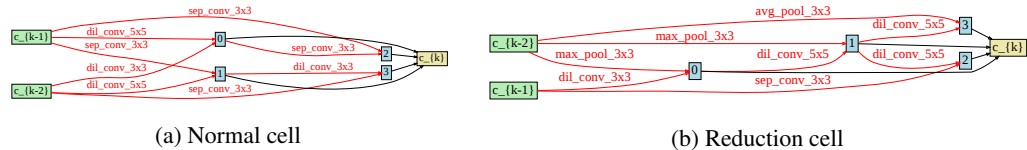

(a) Normal cell

(b) Reduction cell

Figure A15: Cells found by Zero-Cost-PT (`random` discretization order) on the DARTS-S1 space using SVHN.

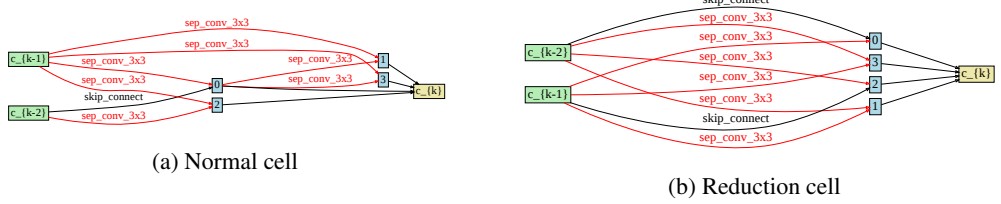

(a) Normal cell

(b) Reduction cell

Figure A16: Cells found by Zero-Cost-PT (`random` discretization order) on the DARTS-S2 space using SVHN.

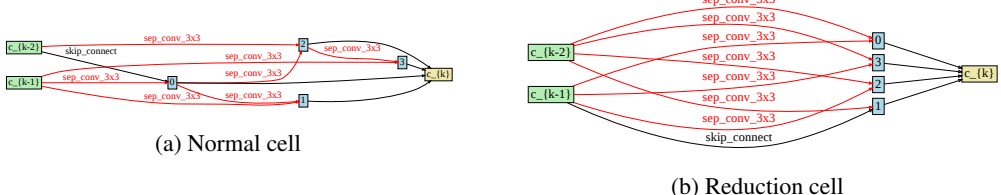

Figure A17: Cells found by Zero-Cost-PT (`random` discretization order) on the DARTS-S3 space using SVHN.

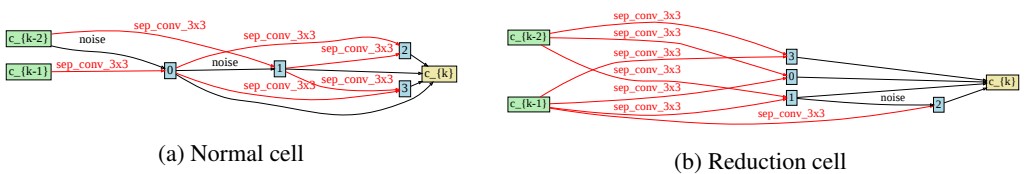

Figure A18: Cells found by Zero-Cost-PT (`random` discretization order) on the DARTS-S4 space using SVHN.

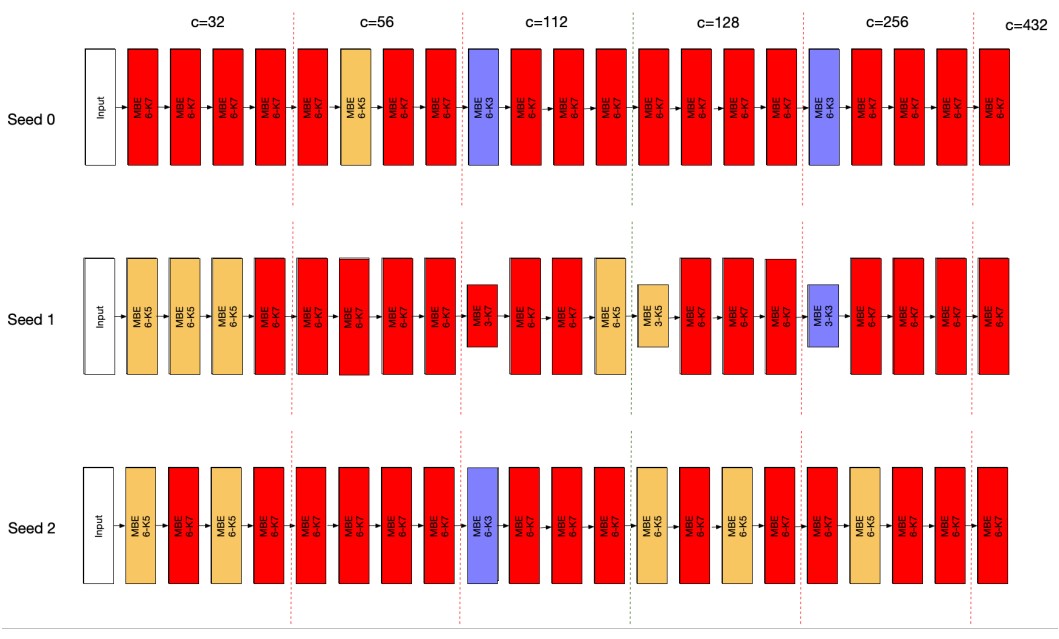

Figure A19: Discovered Architectures in MobileNet-like Search Space

Table A8: Raw values of operation scoring functions at iteration 0 to reproduce Figure 2a.

| | $edge\backslash op$ | none | skip_connect | nor_conv_1x1 | nor_conv_3x3 | avg_pool_3x3 |
|---|---|---|---|---|---|---|
| best-acc | 0 | 94.15 | 94.18 | 94.44 | **94.68** | 93.86 |
| | 1 | 94.24 | 94.16 | 94.49 | **94.68** | 94.09 |
| | 2 | 94.25 | 94.43 | 94.49 | **94.68** | 94.19 |
| | 3 | 94.16 | **94.68** | 94.03 | 94.04 | 93.85 |
| | 4 | 94.29 | 94.18 | 94.56 | **94.68** | 94.23 |
| | 5 | 94.05 | 94.16 | **94.68** | 94.56 | 94.1 |
| avg-acc | 0 | 77.36 | 81.02 | 83.81 | 86.38 | **87.32** |
| | 1 | 80.03 | 83.11 | 85.23 | **85.99** | 81.52 |
| | 2 | 82.9 | 82.44 | 84.05 | **84.49** | 81.98 |
| | 3 | 74.02 | 85.17 | 87.3 | **88.28** | 81.38 |
| | 4 | 80.14 | 83.05 | 85.09 | **85.7** | 81.89 |
| | 5 | 77.61 | 83.43 | 86.18 | **86.95** | 81.74 |
| disc-acc | 0 | **83.27** | 82.24 | 65.0 | 71.76 | 54.31 |
| | 1 | **84.94** | 83.23 | 73.23 | 76.77 | 83.45 |
| | 2 | **83.87** | 83.73 | 77.33 | 76.83 | 83.25 |
| | 3 | 65.77 | **84.44** | 75.82 | 78.68 | 62.7 |
| | 4 | **83.57** | 82.03 | 75.02 | 76.09 | 82.56 |
| | 5 | **83.95** | 82.45 | 66.69 | 71.36 | 80.31 |
| darts-pt[1] | 0 | -85.43 | **-17.02** | -78.13 | -59.09 | -85.34 |
| | 1 | -85.52 | **-36.1** | -84.39 | -80.95 | -85.49 |
| | 2 | -85.51 | -80.29 | -81.86 | **-77.68** | -85.32 |
| | 3 | -85.49 | **-9.86** | -81.79 | -59.18 | -85.48 |
| | 4 | -85.45 | **-51.15** | -78.84 | -64.64 | -85.14 |
| | 5 | -85.54 | **-32.43** | -81.04 | -72.75 | -85.51 |
| disc-zc | 0 | 3331.01 | **3445.49** | 3366.88 | 3437.55 | 3423.18 |
| | 1 | 3429.07 | **3435.75** | 3407.87 | 3434.58 | 3421.44 |
| | 2 | 3428.8 | 3423.36 | **3440.93** | 3437.29 | 3416.89 |
| | 3 | 3408.99 | **3464.05** | 3359.89 | 3382.18 | 3431.81 |
| | 4 | 3433.99 | **3435.57** | 3424.47 | 3431.14 | 3423.15 |
| | 5 | 3434.42 | **3437.66** | 3418.57 | 3397.52 | 3424.17 |
| zc-pt[1] | 0 | -3455.23 | -3449.9 | -3449.54 | **-3441.82** | -3461.18 |
| | 1 | -3452.15 | -3448.7 | -3441.81 | **-3440.65** | -3453.74 |
| | 2 | -3446.52 | -3447.61 | **-3435.46** | -3436.4 | -3449.28 |
| | 3 | -3453.81 | **-3435.99** | -3444.04 | -3445.6 | -3447.07 |
| | 4 | -3451.06 | -3449.8 | -3442.63 | **-3441.13** | -3453.31 |
| | 5 | -3450.97 | -3448.21 | **-3440.8** | -3443.24 | -3452.99 |
| darts | 0 | 0.14 | **0.48** | 0.13 | 0.18 | 0.07 |
| | 1 | 0.12 | **0.55** | 0.11 | 0.12 | 0.09 |
| | 2 | 0.24 | **0.33** | 0.15 | 0.17 | 0.11 |
| | 3 | 0.06 | **0.65** | 0.08 | 0.13 | 0.07 |
| | 4 | 0.12 | **0.48** | 0.13 | 0.17 | 0.1 |
| | 5 | 0.16 | **0.49** | 0.12 | 0.14 | 0.09 |
| tenas | 0 | -38.5 | -48.0 | -31.0 | **-6.0** | -37.5 |
| | 1 | **-7.0** | -55.0 | -10.0 | -15.0 | -39.0 |
| | 2 | -31.5 | **-10.0** | -30.0 | -16.5 | -36.5 |
| | 3 | -34.0 | -44.0 | -53.5 | **-23.0** | -30.0 |
| | 4 | -32.0 | -32.5 | -36.5 | **-32.0** | -52.0 |
| | 5 | -38.5 | **-16.0** | -20.0 | -17.0 | -27.5 |

[1] Lower is better so we add a negative sign to *-pt scores.

