# OpenReview forum: "Zero-Cost Operation Scoring in Differentiable Architecture Search"
_ICLR.cc/2022/Conference — ICLR 2022 Submitted_

### Official Review · Reviewer_86cZ · 2021-10-16

**Correctness:** 3
**Technical Novelty And Significance:** 2
**Empirical Novelty And Significance:** 2
**Recommendation:** 5
**Confidence:** 4

**Main Review:**

Strength:
This work studied different training-free proxies and leveraged perturbation-based methods for NAS.

Weakness:
My main concern is the novelty of this work. Specifically,
1. This work did not propose any new proxies for evaluating networks performance. If I understand correctly, the proxies listed in Table 2 are all from previous works. The authors did not claim any novelty in these training-free proxies either. Although the authors mentioned "opportunity to use a new class of training-free proxies" in Sec. 3.1, there is no concrete proposal in this work.
2. The operation sorting method (Eq. 5) is also similar to the perturbation method proposed by [1]. The main difference is that the authors adopt this perturbation at supernet's initialization.

[1] Rethinking architecture selection in differentiable NAS.

**Summary Of The Paper:**

This work combines training-free proxies and perturbation-based subnetwork evaluation to achieve efficient NAS. The author also conducted comprehensive study of training-free proxies under different training iterations.

**Summary Of The Review:**

In addition to the novelty issue mentioned above, some results and description of this work also makes me a bit confusing. For example, when demonstrating Figure 2, the author did not mention which concrete training-free proxy they used for $S$ in Eq. 5. Also, the purpose of "retrain for 5 epochs" is unclear to me. Why does the author want to conduct this experiment given the "zero-cost" motivation in this work?

---

> ### Author Response · Authors · 2021-11-20
> **Thank you for your review.**
>
> We thank the reviewer for the helpful comments on our work. We hope to have clarified both of your concerns satisfactorily below. Please do let us know if you see any further issues in the paper that are unclear or need to be addressed.
>
> > Proposing new training-free proxies.
>
> We agree with the reviewer that our paper does not propose new proxies for evaluating network performance, and indeed we do not claim novelty in the training-free proxies themselves considered in our paper either. In Sec. 3.1, what we meant by “opportunity to use a new class of training-free proxies” is that those proxies, although have been shown to perform well in non-supernet based NAS (e.g. [1] [2]), have not been studied under differentiable NAS. This paper exploits exactly this opportunity, and proposes:
>
> - A novel formulation of operation scoring functions is differentiable NAS and comprehensive analysis of those functions in both initial and progressive settings (Section 3). To the best of our knowledge, this is previously not investigated in the differentiable NAS context. By analyzing operation scoring in a clearly defined (Section 3.1) and reproducible (Section 3.2) way, we improve our understanding of operation scoring in differentiable NAS, in that the zero-cost metrics actually work very well in selecting the best operation at each edge on a supernet. As shown in Figure 1, this is only true if zero-cost metrics are used with the perturbation paradigm, showcasing limitations of the existing approaches presented in [1, 2]. We argue that such an analysis is the only way to test which method (or _proxy_) is most closely correlated to the real objective of selecting the best operation on each edge.
>
> - A much more efficient NAS algorithm Zero-Cost-PT (Section 4), which leverages the strong performing zero-cost proxies and operation perturbation paradigm in the differentiable NAS settings. We conduct thorough empirical evaluation on both small (NAS-Bench-201), large (DARTS) and other (DARTS S1-S4, MobileNet-like) search spaces. This analysis doesn’t exist in the existing literature concerning zero-cost proxies and we believe there is novelty in presenting these results. Overall, our Zero-Cost-PT outperforms most baselines with a significantly faster search time.
>
> It is also worth pointing out that both our analysis and the proposed Zero-Cost-PT approach are general and can be extended to new classes of training-free proxies that may emerge in the future. This is evident by our ablation study (Sec. 4.2), in which we consider a variety of existing zero-cost proxies in our Zero-Cost-PT approach, and compare the performance of those variants on NAS-Bench-201 (Table. 2).
>
>
> > Novelty in operation scoring.
>
> We thank the reviewer for this comment. We agree that our work uses the same idea of perturbation as in DARTS-PT [3]. However as discussed above, combining the perturbation paradigm and zero-cost metrics with a thorough empirical evaluation is part of our contribution (Section 4).  The other part is the formalization and quantitative analysis of operation scoring to distinguish between differentiable NAS methods in Section 3. As stated earlier, the results of Section 3 are what motivated our Zero-Cost-PT in Section 4. We believe that these two pieces together constitute a significant contribution to the differentiable NAS literature. We are able to better understand the operation scoring proxies used in existing methods, and we presented a low-cost algorithm to further improve upon these existing methods in a number of popular NAS search spaces.
>
> > Clarification on some results and discussion.
>
> We thank the reviewer for this set of comments, and please kindly find our clarifications below:
>
> - In Figure. 2, the zero-cost proxy we use for `disc-zc` and `zc-pt` is `nwot` (in Equation. 5), as mentioned in the later ablation study. We will add a note to the captain of the figure as well as the text.
>
> - Throughout our paper (e.g. in Section 3.2), “retrain for 5 epochs” refers to `disc-acc` and `darts-pt`, which is not our approach but taken from DARTS-PT [3]. Our Zero-Cost-PT doesn’t require this step of retraining between operation selections, and thus is far more computationally efficient than DARTS-PT.
>
>
>
> [1] Mohamed S Abdelfattah, Abhinav Mehrotra, Łukasz Dudziak, and Nicholas Donald Lane. Zero-cost proxies for lightweight NAS. In International Conference on Learning Representations (ICLR), 2021.
>
> [2] Joseph Mellor, Jack Turner, Amos Storkey, and Elliot J. Crowley. Neural architecture search without training. In International Conference on Machine Learning (ICML), 2021.
>
> [3] Ruochen Wang, Minhao Cheng, Xiangning Chen, Xiaocheng Tang, and Cho-Jui Hsieh.  Re-thinking architecture selection in differentiable NAS. In International Conference on Learning Representations (ICLR), 2021.

---

> > ### Comment · Reviewer_86cZ · 2021-11-22
> > **Thanks for the response!**
> >
> > Dear authors,
> >
> > I appreciate the detailed response to my concerns. However, after reading I still believe the contribution of this work is incremental. I appreciate the author's hard work and I believe it takes a huge amount of effort in making zero-cost proxies work in differentiable NAS. But I could not learn anything new from this work: 1) no new proxies introduced; 2) I am not convinced why using zero-cost proxies for **differentiable NAS** is a must-have, why not instead of using them in Reinforcement Learning or Evolution Search? Moreover, I believe there could be some zero-cost proxies that are hard to differentiate, which indicates that differentiable NAS may not always be the best choice when we want to leverage zero-cost proxies to accelerate NAS.
> >
> > Therefore, I choose to keep my score.

---

> > > ### Author Response · Authors · 2021-11-22
> > > **Clarification on differentiability**
> > >
> > > Thank you for your prompt response.
> > >
> > > > No new proxies introduced
> > >
> > > As in our previous responses to the reviewer’s comments, indeed we do not introduce any new proxies, but we use the proxies in a novel way to score operations, and we investigate exactly how to do that either through discretization or perturbation — we consider this to be our main contribution as opposed to coming up with a new NAS proxy. Our goal was to extend zero-cost NAS to larger search spaces that are typical in differentiable architecture search, and we wanted to demonstrate a search time speedup compared to existing approaches.
> > >
> > > > Using zero-cost proxies for differentiable NAS vs. in Reinforcement Learning or Evolution Search.
> > >
> > > We would like to clarify one thing that might have been miscommunicated by us due to imprecise wording. Our method *does not* require a differentiable setting.
> > > Specifically, we never optimize a network's architecture with gradient descent-like methods and we do not need to calculate derivatives of any zero-cost metrics (as rightly noted by the reviewer, for some of them this might be hard or even impossible), although some of these metrics do require us to calculate standard gradient w.r.t. a model's parameters (notably, naswot, which we mostly use, only requires a single forward pass so we don't even need a differentiable loss in theory).
> > >
> > > What we meant by saying that our method focuses on "differentiable NAS" was that we focus on utilizing zero-cost proxies in a supernet-based search.
> > > This is motivated by the fact that search space size grows exponentially and it quickly becomes inefficient to simply try out different models and score them with zero-cost proxies, which was suggested in the previous works and which we show achieves worse results than our method in Section A.6.2 of the Appendix.
> > > Instead, it is necessary to employ some kind of divide-and-conquer method that would allow us to build a globally-good solution from local optimizations, which is exactly what we attempt in our work.
> > > In this context, supernet-based search provides us with opportunities to do that (Section 3.1) and we chose to focus on it because of that.
> > >
> > > To answer the reviewer's question about RL/Evo search with zero-cost proxies, we would like to mention that in the context of multi-trial NAS this has already been done by Abdelfattah et al. in [1]. The problem with those methods is that they still require training of multiple models and hence are significantly more costly.
> > > Approaching the problem with a supernet-based solution allows us to alleviate this limitation but also makes the adoption of RL/Evo algorithms much more challenging as we do not have any feedback mechanism which would be cheap enough to drive those methods (even training a single model might be too expensive).
> > > However, there might be an opportunity to use RL/Evo methods in our architecture proposal phase, as explained in our response to reviewer podk, which we leave for future work.
> > >
> > > We hope that this clarifies things in case our initial wording caused some confusion -- we will update our paper to avoid suggesting that our method requires a differentiable setting.
> > > We would like to thank the reviewer again for their valuable feedback.
> > >
> > >
> > >
> > > [1] Mohamed S Abdelfattah, Abhinav Mehrotra, Łukasz Dudziak, and Nicholas Donald Lane. Zero-cost proxies for lightweight NAS. In International Conference on Learning Representations (ICLR), 2021.

---

### Official Review · Reviewer_q4ge · 2021-10-30

**Correctness:** 3
**Technical Novelty And Significance:** 2
**Empirical Novelty And Significance:** 3
**Recommendation:** 5
**Confidence:** 4

**Main Review:**

Pros:
1. The paper is well-written and easy to follow. However, analyze part is slightly too long however the paper is in general in good shape.
2. The experiment results look promising and interesting. It could achieve better accuracy across several benchmarks and some spaces, which could be useful to the community.

Cons:
1. The paper's novelty is quite limited. It simply combines the differentiable neural architecture search with the one-shot one. Although it has been shown and addressed across the paper the zero-cost proxies are quite useful, however, it lacks insight on how it is correlated with the final performance ranking.
2. At the same time, all the zero-cost proxies are proposed by other works and there is limited understanding of why some specific proxies are clearly better than others in the differentiable setting but not in the one-shot setting.
3. Although the paper formalizes the problem into perturbation and discretion, I find only discretization is used in the final experiments, where the formalized framework seems not that important.

**Summary Of The Paper:**

The paper proposes to use the zero-cost proxy in the operation selection in differentiable neural architecture search. Specifically, the paper formalizes the selection procedure into perturbation and discretization. By introducing the zero-cost proxy, they first find the proxy could yield a better ranking compared to both Darts and Darts-PT in the Nas-Bench 201. They also experiment with different proxies proposed by previous work and find they all perform better than the Darts-PT. In the Darts-CNN and S1-S4 experiments, they select one of the best-performed proxies and find it could yield better accuracy than other baselines.

**Summary Of The Review:**

Although the paper has shown us the zero-cost proxy has great power in operation search in the differentiable neural architecture search, the paper stills lack insight on analyzing how it works. Therefore, I rate it as a borderline paper.

---

> ### Author Response · Authors · 2021-11-20
> **Thank you for your feedback.**
>
> We thank the reviewer for the comments on our work. In the following we provide our responses and additional investigations conducted with respect to the reviewer’s comments, and hope to clarify the concerns satisfactorily. Please do let us know if you see any further issues in the paper that are unclear or need to be addressed.
>
> > Insight on how zero-cost proxies are correlated with the final performance ranking.
>
> We thank the reviewer for this comment. However, the main focus of our work is to investigate the effectiveness of **zero-cost operation scoring** in the context of differentiable architecture search, not the final performance ranking of architectures evaluated with the zero-cost proxies. Indeed there are prior works (e.g. [1], [2]) that use zero-cost proxies to score the final performance of searched architectures and study their correlations with final accuracies. In addition to consistent strong performance compared to those works, our work is orthogonal in the sense that we look at how operations are scored at each edge in a supernet using metrics including the zero-cost proxies, and compare different operation scoring functions to the oracle. We believe this is one of the key novel contributions of our paper, and hasn’t been done before in the context of differentiable NAS. We will clarify this in Section 3, and add more discussion in Section 4 and 5 when referring to the existing work on zero-cost proxies.
>
> > Understanding of the zero-cost proxies.
>
> We agree that our paper considers the existing zero-cost proxies. However, as we mentioned in the paper and responses to other comments, the novelty of our work is not in proposing new zero-cost proxies, but a novel formulation and systematic study on how these proxies can be used as operation scoring functions in differentiable NAS settings (Section 3). In particular, we conduct ablation studies by comparing different zero-cost proxies side-by-side, and based on the empirical results, we design our NAS algorithm (Zero-Cost-PT) using the best performing metrics (Section 4). Although existing work [1] has studied different zero-cost proxies in traditional NAS context, our work complements it by comparing them in the differentiable context, and showing that those metrics, when combined with the perturbation-based NAS paradigm (our Zero-Cost-PT), could become promising yet efficient proxies to the actual model accuracies. In addition, we also show in the appendix (A.6.2) that a simple application of zero-cost NAS [1] on DARTS CNN search space, even with the best metric (`nwot`), falls visibly behind the proposed Zero-Cost-PT.
>
> We are not sure what the reviewer meant by “some specific proxies are clearly better than others in the differentiable setting but not in the one-shot setting”, and would like the reviewer to kindly clarify this comment for us.
>
> > Formalization of perturbation and discretization and its relation to the final experiments.
>
> We thank the reviewers for pointing this out. Indeed in Section 3 we formulate both perturbation and discretization paradigms for operation scoring in differentiable NAS context, and further study the properties of different operation scoring metrics based on both discretization (`disc-acc` and `disc-zc`) and perturbation (`darts-pt` and `zc-pt`). Through our analysis, we find that discretization based operation scoring functions are either inversely (`disc-acc`) or weakly ( `disc-zc`) correlated with the ground truth (the oracle `best-acc`) in both initial (Section 3.2.1) and progressive (Section 3.2.2) settings. Therefore we conclude that perturbation is generally a better scoring paradigm than discretization, which is adopted in our NAS algorithm in Section 4 as well as experiments reported in Section 5.
>
> To complete our study and address the reviewer’s comments, we launched additional experiments, using discretization instead of perturbation in our NAS algorithm. In particular, we use the exact same hyper-parameters with the proposed Zero-Cost-PT (`N=10`, `V=100`, `nwot`), but use `disc-zc` as the operation scoring function instead of `zc-pt`. We denote this baseline as Zero-Cost-DISC, and the results on NAS-Bench-201 are:
>
> | **Method* |**CIFAR-10** |**CIFAR-100**|**ImageNet-16**|
> | -: | :--: | :--: | :--: |
> | Zero-Cost-DISC |6.22$\pm$0.84 | 28.18$\pm$2.01 | 55.14$\pm$1.77 |
> | Zero-Cost-PT |5.97$\pm$0.17 |27.47$\pm$0.28 |53.82$\pm$0.77 |
>
> We see that discretization (Zero-Cost-DISC) results in inferior performance compared to the proposed perturbation-based approach (Zero-Cost-PT) on all datasets, confirming our previous analysis on their correlations with the oracle metric. We hope this is enough to address the concerns from the reviewer. We are going to add the above results and relevant discussion in the appendix.
>
>
> [1] M. S. Abdelfattah, et al. Zero-cost proxies for lightweight NAS. ICLR'21
>
> [2] J. Mellor, et al. Neural architecture search without training. ICML'21.

---

> > ### Comment · Reviewer_q4ge · 2021-11-30
> > **Responses to authors**
> >
> > Thanks for the detailed feedback. After reading the rebuttal, I still have concerns about incremental novelty. The results in table 2 show  TE-NAS is good to expect imagined results, which means there might not be a golden score to select. In this sense, how to select which proxy becomes a new problem and clearly there is no insight for us to learn how to select the best zero-order proxy. As there is no clear sign of how to select a zero-order proxy, the solution would be trying every proxy in every space to get a better performance, which is not practical in my opinion. I will keep my score.

---

> > > ### Author Response · Authors · 2021-12-01
> > > **Comparison to TE-NAS and what proxy to use**
> > >
> > > We thank the reviewer for this comment. We agree that, in general, it is unlikely to have a universal "golden score" that performs well across all datasets for all tasks. However, in the context of our work, we disagree with the statement that "there is no clear sign of how to select a zero-order proxy".
> > > Before we provide more details, it is worth mentioning that TE-NAS is a NAS approach that combines both a zero-cost proxy (NTK + Linear regions) and a discretization policy that uses pruning (where we use perturbations). Since the reviewer based their comment on the TE-NAS results, we will comment on comparison to TE-NAS only, but similar argumentation would still hold for other proxies/algorithms.
> > >
> > > We analyze correlation of the proxy used in TE-NAS in Section 3 and show that it correlates worse than naswot (Figure 2) and, consequently, yields worse results when used to discretize edges independently (Table 1).
> > > When the TE-NAS proxy is used with the pruning-based discretization, we get results shown in Section 4 onwards, e.g., in Table 2 that the reviewer mentioned. The results are on par with our proposed ZC-PT on the DARTS CNN CIFAR-10 task (+0.01 pp but ~3x slower) and fall behind on others, with imagenet results being quite significantly worse (-1.8 pp).
> > > Overall, our end-to-end results suggest that the naswot with perturbation paradigm is a clear winner, and ablations/analysis show better properties of the naswot metric compared to the NTK+Linear regions from TE-NAS.
> > >
> > > Therefore, we are not sure if the reviewer's criticism that we'd need to try every proxy in every space is justified - even in our paper we only tested different proxies on NB2 (Table 2, first part) and showed that it is enough to take the best performing one from this experiment to achieve satisfactory results on many different tasks (DARTS-CIFAR10, DARTS-ImageNet, S1-S3, MobileNet-like). Instead of focusing our work on "what" metric to use, we show that it is more important to answer the question "how" to use a selected metric. In this context, we would like to notice that our strong empirical results are not simply a result of using a well-performing zero-cost metric, but it was necessary to also carefully design a searching algorithm on top of that (one of our contributions, for more details about the design challenges please see our response to reviewer podk) - as evidenced by the results comparing standard NASWOT and Zero-Cost-Disc to our Zero-Cost-PT.
> > >
> > > We would like to thank the reviewer again for their valuable feedback and we hope that our response clarifies some of the concerns that the reviewer had.

---

### Official Review · Reviewer_G1Xa · 2021-11-02

**Correctness:** 3
**Technical Novelty And Significance:** 3
**Empirical Novelty And Significance:** 3
**Recommendation:** 6
**Confidence:** 3

**Main Review:**

Pros:
This paper introduces the lightweight operation scoring based on zero-cost proxies, which empirically outperform existing operation scoring functions. The experiments show that the perturbation is more effective than discretization and propose Zero-Cost-PT. Extensive experiments empirically show that the proposed method outperforms the best available differentiable architecture search in terms of searching time and accuracy.

Cons:
The zero-cost proxies method has been used in the previous work. This work only borrows it into differentiable architecture search, which is only a combination. There exist some typos and it is a little difficult to understand Figure 1. Some annotations can be moved to the caption.


**Summary Of The Paper:**

In this paper, the authors introduce new training-free proxies to the context of differentiable NAS, which can speed up the search process while improving accuracy. Further, the authors propose, evaluate and compare perturbation-based zero-cost operation scoring (Zero-Cost-PT) for differentiable NAS. Extensive experiments empirically show the effectiveness of Zero-Cost-PT in six search spaces and 3 datasets.

**Summary Of The Review:**

Generally speaking, I think it is incremental work. Although there is a lack of theoretical contributions, a large number of experiments provide empirical contributions. It is difficult for me to put forward some new opinions about these large numbers of experiments. I give the above acceptance threshold, but reject is OK.

---

> ### Author Response · Authors · 2021-11-20
> **Thank you for your positive feedback.**
>
> We appreciate the reviewer for the helpful comments on our work.
>
> > “The zero-cost proxies method has been used in the previous work. This work only borrows it into differentiable architecture search, which is only a combination.”
>
> We agree that the zero-cost proxies methods considered in our paper have been used in previous work. However, as pointed out by Reviewer podk , “It is a natural idea to combine… but also not trivial to do it well.” We would like to highlight the major contributions made in this paper:
>
> -  We propose a novel formulation of operation scoring and conduct a novel quantitative analysis of these functions (Section 3). This is the **first time** differentiable NAS operation scoring methods are analyzed side-by-side in detail, other than just comparing the final model accuracy. Our novelty is in the formal definition of operation scoring functions and, thanks to NAS-Bench-201, we are able to exactly quantify how well each operation scoring function performs compared to an oracle/perfect operation scoring function, in both initial and progressive settings.
> - Motivated by our analysis on the operation scoring functions, we further propose Zero-Cost-PT (Section 4), which leverages the strong performing zero-cost proxies and operation perturbation paradigm in the differentiable NAS settings, combining the best of both worlds. We also provide thorough empirical evaluation on both small (NAS-Bench-201), large (DARTS) and other (DARTS S1-S4) search spaces. This analysis doesn’t exist in the literature concerning zero-cost proxies and we believe there is novelty in presenting these results. Finally, our Zero-Cost-PT outperforms most baselines with a significantly faster search time.
>
> > Typos and unclarity in Figure 1.
>
> We thank the reviewer for pointing this out. We will update Figure 1 to clear some of the annotations, and smooth out the text in the caption.
>
> We hope to have clarified both of your concerns satisfactorily. Please do let us know if you see any further issues in the paper that are unclear or need to be addressed.

---

> > ### Comment · Reviewer_G1Xa · 2021-11-29
> > **Response to Authors**
> >
> > Thanks for your feedback, I have read the authors' rebuttal and other reviewers' comments. I still think my review is reasonable, this manuscript is near the borderline, so I keep my score.

---

### Official Review · Reviewer_podk · 2021-11-02

**Correctness:** 3
**Technical Novelty And Significance:** 3
**Empirical Novelty And Significance:** 3
**Recommendation:** 8
**Confidence:** 4

**Details Of Ethics Concerns:**

None.

**Main Review:**

## Strengths
- This is a great combination of two popular areas: zero-cost proxies, and differentiable NAS via perturbation-based operation scoring. It is a natural idea to combine these two things, but also not trivial to do it well.
- These ideas can have a high impact.
- It seems that they give new insights for the celebrated “Rethinking Architecture Selection in Differentiable NAS” paper, including more proof of why DARTS-PT does well, and a refutation of the claim that disc-acc does well.
- The appendix is pretty good and has a lot of information for reproducibility (but for code, see weaknesses).
- Sections 3.1 and 3.2 are nicely done and make sense, and Fig 2 in particular is very interesting.

## Weaknesses
- Over-reliance on NAS-Bench-201. Almost all of Sections 3 and 4 focus on NAS-Bench-201. But this is the smallest search space, and in particular, it is known that zero-cost proxies do well on NAS-Bench-201 and bad on larger search spaces (the authors mention this as well). Although the authors also test their full algorithm on the DARTS search space in Section 5, it would strengthen the paper to do the analysis of Sections 3 and 4 on at least one other search space. NAS-Bench-1shot1 is the best option because it is size 360k and can run one-shot algorithms. TransNAS-Bench-101 can also run one-shot, and it is smaller but would add diversity. Finally, it might be possible to run some of the experiments using the 60k trained architectures from NAS-Bench-301).
- Better ablation between the search and the validation stage. Sections 3 and 4 motivate the search part of the algorithm, but adding in the validation stage makes it less clear what leads to strong performance, especially on DARTS where the search step was never run in isolation. I think it makes it less apples-to-apples comparisons, e.g. because DARTS, DARTS-PT etc do not have the final validation stage. The authors did have an ablation, but it is only on NAS-Bench-201 and does not cover search only ("validation only" is basically NASWOT). In order to have a better comparison between ZC-PT and DARTS-PT, the authors could run ZC-PT once with no validation stage. Or, run ZC-PT and DARTS-PT both V times and do the validation stage for both of them. For Tables 2, 4, 5. As a result, I think the claim in the intro that "Our novel Zero-Cost-PT improves searching time and accuracy compared to the best available differentiable architecture search for many search space sizes, including very large ones." is misleading.
- The authors claim in Section 7 that they released their code in the supplementary material, but unfortunately there was no supplementary material submitted. If the authors provide the code during the rebuttal period, I will feel more positive about the paper (for example, anonymous github).


**Summary Of The Paper:**

Differentiable neural architecture search (NAS), and more recently, perturbation-based operation selection in differentiable NAS, is a popular recent area of study in NAS. In parallel, zero-cost proxies are also gaining in popularity in NAS. In this paper, the authors combine the insights from perturbation-based operation selection and zero-cost proxies, to create a new method that sees substantial time speedups. Specifically, they formalize "operation scoring", and use zero-cost proxies to score operations with the perturbation paradigm. This leads them to a new NAS method (Zero-Cost-PT) that outperforms existing methods. They evaluate their proposed algorithm on the DARTS search space (and its subsets) and NAS-Bench-201, showing that it can achieve up to 40x speedups.

**Summary Of The Review:**

This paper combines two popular techniques in a nice way. I think the ideas in this paper can be pretty impactful. I do have three weaknesses about reliance on NAS-Bench-201, a fairer experimental setup, and code release, so I will give a weak accept.

---

> ### Author Response · Authors · 2021-11-20
> **Thank you for your positive feedback. (2/2)**
>
> *-- (continuation from the previous post) --*
>
> Indeed, our paper currently lacks analysis of how our method behaves on the DARTS CNN space with N=1, therefore we have run relevant experiments and present those results in the table below.
>
> | **Searching Seed** |  **Test Error (%)** (4 trainings)     |
> |:----------------:|:---------------:|
> |              0 |   2.72 / 2.55 / 2.83 / 2.71 |
> |              1 |   3.25 / 3.26 / 3.28 / 3.20 |
> |              2 |   2.59 / 2.84 / 2.55 / 2.79 |
> |              3 |   2.43 / 2.77 / 2.52 / 2.66 |
> | Average   | 2.81$\pm$0.29 |
> | Best         | 2.43       |
>
> As can be seen, the average performance is affected quite significantly. However, the best model still happens to be on-par with our main results. This suggests what has already been mentioned in the submitted version - that searching phase alone tends to “find” many different architectures depending on, broadly speaking, random seed, and this randomness is especially visible in the case of random edge ordering (Figure 3a). While the high variance might seem undesired at first, we empirically observe that the higher exploration resulting from it is beneficial for finding some very good models - e.g., global-op-iter discretization order tends to be less sensitive to random seed as it takes away one degree of randomness (edge order), producing more stable results on average, but at the same time limiting its ability to maximize performance of the best model found (Table 4). In order to maximize performance of our method, we balance exploration (higher N + random edge order) and exploitation (higher V) in the searching and validation phase respectively. Admittedly, the interplay between those two phases is crucial for our method. To further showcase how the validation phase complements the searching phase, we run additional ablations on the DARTS CNN space with N=10 and V={1,10,100}, the results are shown in the table below.
>
> | **Searching Seed** | `V` = 1 **Test Error (%)** | `V`  = 10 **Test Error (%)** | `V`  = 100 **Test Error (%)** |
> |:----------------:|:-------------------:|:---------------------:|:----------------------:|
> |              0 | 3.08 / 3.16 / 3.06 / 2.96 | 3.08 / 3.16 / 3.06 / 2.96 | 2.86 / 2.97 / 2.77 / 2.82 |
> |              1 | 2.74 / 2.91 / 2.92 / 2.87 | 2.74 / 2.91 / 2.92 / 2.87 | 2.51 / 2.47 / 2.56 / 2.43 |
> |              2 | 2.96 / 3.10 / 3.06 / 2.90 | 2.77 / 2.71 / 2.65 / 2.76 | 2.74 / 2.73 / 2.54 / 2.62 |
> |              3 | 2.86 / 2.85 / 2.85 / 2.65 | 2.83 / 3.00 / 2.82 / 2.87 | 2.43 / 2.77 / 2.52 / 2.64 |
> | Average        | 2.93$\pm$0.14        | 2.88$\pm$0.14          | 2.64$\pm$0.16           |
> | Best         | 2.65              | 2.65                | 2.43                 |
>
> The results are consistent with what is shown in the paper, that is: higher V produces better results on average but does not affect the best case that much (the best model is still upper-bounded by what was found with N=1),
>
> We consider those extra results to be completing the picture. Regarding comparison to DARTS-PT and whether it is fair or not, we are not sure what exactly are the reviewer’s concerns. It is true that DARTS-PT does not have a validation phase but the cost of running it multiple times, considering that it requires training of a supernet, would be too expensive to suggest something like that in our opinion (please note that in our case during validation a network is still untrained). For example, accuracy-wise the best-case scenario for running DARTS-PT multiple times is already included, e.g., in Table 4’s “best” column, while its cost would increase to 9.2 (assuming full training for 1.5 GPU days to validate each network). At the same time, the increase in cost would leave some extra room for our ZC-PT - in particular, since our method is cheaper per single run we can also run it multiple times (end-to-end) and achieve the result showed in the “best” column as well, with total cost of 6.07. We would argue that this is sufficient for us to make a claim that our method is able to find accurate models faster. We are happy to further discuss this aspect if the reviewer remains unconvinced.
>
> As previously, all the above additional results and relevant discussion will be added to the Appendix of our paper.
>
> > Code release.
>
> We thank the reviewer for pointing this out, a link to our repo can be found in the previous message (addressed to all reviewers) that we posted a few days ago.
>
>
> We appreciate the time and effort that the reviewer has shown to help us improve our work. We hope our responses above could have addressed all your concerns, and please do let us know if you see any further issues in the paper that are unclear or need to be addressed.
>
>
> [1] Arber Zela, Julien Siems, Frank Hutter. NAS-Bench-1Shot1: Benchmarking and Dissecting One-shot Neural Architecture Search. In International Conference on Learning Representations (ICLR 2020).

---

> > ### Comment · Reviewer_podk · 2021-11-21
> > **Thanks for the reply**
> >
> > I thank the authors for the work they did in preparing the rebuttals. The experiments on nas-bench-1shot1 particularly improve the paper, and releasing the code. I also like how the authors changed “search” to “architecture proposal” now. It makes it more clear that architecture proposal is the main algorithmic contribution of this work, since the validation stage is from NASWOT.
> >
> > It is good that the authors have clarified that the interplay between the architecture proposal phase and the validation phase is crucial for their method. And the authors are honest that their method does not work quite as well on nas-bench-1shot1 as on nas-bench-201, which is to be expected because zero-cost proxies work best on smaller search spaces.
> >
> > Eventually, the authors could put the main summaries of those new points into the first 9 pages.
> >
> > Are there any other “architecture proposal” methods the authors can compare their method to (with the same validation phase)? “Random” as a baseline is already there, since that is NASWOT. And I agree now that DARTS-PT etc are much more costly.
> >
> > My rating for this paper is now closer to “8” than “6” so I am increasing my rating (I think it’s a 7, but we don’t have that option).

---

> > > ### Author Response · Authors · 2021-11-22
> > > **Other architecture proposal methods**
> > >
> > > Thank you for your prompt response and support for our work.
> > >
> > > > Other “architecture proposal” methods to compare
> > >
> > > We are not aware of any existing approach to propose architectures that would be sufficiently cheap to use it within our zero-cost method without increasing its cost significantly. However, we anticipate that it could be possible in general to directly search for models that achieve high zero-cost metric - we then could utilize some of the existing, efficient algorithms that are normally used with multi-trial NAS, such as RL, evolution or predictors. This approach however might overfit to the zero-cost proxy and also might not be as fast as our ZeroCost-PT as most of those methods scale worse with the search space size (they are not divide-and-conquer and/or perform training of a controller/predictor that can be costly). Therefore it’s not immediately obvious if there’s benefit in taking this road. We leave this question for future work.
> > >
> > > We would like to thank the reviewer again for their valuable feedback.

---

> ### Author Response · Authors · 2021-11-20
> **Thank you for your positive feedback. (1/2)**
>
> We thank the reviewer for the helpful feedback on our work. Please find our responses to the three comments as follows:
>
> > Over-reliance on NAS-Bench-201.
>
> We agree with the reviewer that NAS-Bench-201 is a small search space, and therefore in our paper we also evaluate our Zero-Cost-PT approach thoroughly on the much bigger DARTS CNN space, and four of the DARTS subspaces to study its robustness (Section 4). Additionally, as suggested by the reviewer, to strengthen our paper we have conducted the following additional experiments on the NAS-Bench-1shot1 benchmark [1].
>
> *(Correlation scores can be seen at:
> https://ibb.co/jgw3Y3b)*
>
> We first extend our correlation analysis from Section 3. We compare to DARTS as it is already available in the NB1shot1 codebase. The results show that DARTS is surprisingly well-correlated to both best and avg accuracy, in some cases even better than our proposed ZC-PT. We believe that this is because the search space does not contain skip connections and overall is rather unusual compared to others used with differentiable NAS, so it is possible that it constitutes an edge-case where DARTS performs relatively well.
>
> When it comes to NAS results, we can see that DARTS retains its high performance. On the other hand, our ZC-PT achieves worse average performance, with noticeable variance (especially on the largest Space 3), but is also able to find better models in the best-case scenario. We would like to notice that unlike DARTS or NB2, NB1shot1 supernet contains architectural parameters associated with entire connections between cells (alpha and gamma in the NB1shot1 paper), additionally to the standard ones associated with candidate operations in a single layer (beta in the NB1shot1 paper). What it means for our method is that for those parameters we no longer perturb a single edge of a supernet but rather the entire path, making the setting for our algorithm noticeably different. Further extending our method from operation scoring to path scoring is a very relevant goal for future work.
>
> | **Space** |  | **ZC-PT** | **DARTS** |
> | :--- | :---: | :---: | :---: |
> | 1 | Avg. | 6.76$\pm$1.05 | 6.67$\pm$0.08 |
> |  | Best | 5.45              | 6.44 |
> | 2 | Avg. |6.84$\pm$0.40 | 6.68$\pm$0.36 |
> |  | Best | 6.14             | 6.24 |
> | 3 | Avg. | 7.45$\pm$0.94 | 6.64$\pm$0.12 |
> |  | Best | 6.03              | 6.5 |
> (values are test error %)
>
> We are working on adding DARTS-PT to the results on NB1shot1 - we will add a separate comment when it is ready but please be advised that it might happen a bit later during the discussion period.
>
> All the above additional results and relevant discussion will be added to the appendix of our paper, with relevant pointers in the main text.
>
> > Better ablation between the search and the validation stage.
>
> We agree with the reviewer that due to the coupling of the two phases of our algorithm in Section 5 it might be unclear how different parts contribute to the final outcome. We attempt to clarify that in our response below.
>
> First, for the sake of clarity, our method is related to NASWOT in the following ways: 1) NASWOT is achieved when we generate N subnetworks randomly and then follow with “validation phase” - the search phase is still needed because without it we wouldn't have any networks to validate, although in standard NASWOT this is done very simply; 2) consequently, our method extends NASWOT by including more sophisticated search phase based on the combination of zero-cost metrics and perturbations. Perhaps a better name for “search phase” would be “architecture proposal phase”. Architectures are then evaluated in a lightweight manner during the validation phase to come up with a single outcome of a search.
>
> *-- (continued in the next post) --*

---

### Decision · Program_Chairs · 2022-01-20

**Decision:**

Reject

**Comment:**

This paper received scores of 5,5,6,8. The reviewer giving a score of 8 stated that they would've given a 7, but that that is not an option in the system. The other reviewer giving an acceptance scores mentioned that they would also be OK with a rejection. The details of the assessment are thus less enthusiastic than could be assumed with an overall average score of 6. I am therefore weakly recommending rejection.

The main criticisms of the reviewers are lack of novelty, lack of deeper analyses that really provide insights into why zero-cost operation scoring works, and lack of the number of NAS benchmarks tested. Out of these, personally, I would not criticize a lack of novelty, since it is not trivial to put together zero-cost and one-shot methods and the results appear promising.
However, even the most positive reviewer criticized that the work focuses on NAS-Bench-201 heavily (which is particularly problematic given that NAS-Bench-201 uses a fixed wiring and only allows the choice of operations; this may make the proposed method particularly applicable). During the rebuttal, the authors added NAS-Bench-1shot, which is a very good step, but the proposed technique does not actually work as well there. While this may be due to the special nature of operations in the nodes rather than in the edges for NAS-Bench-1shot1, for a revision, it would be good to add additional experiments on further NAS benchmarks in order to allow for a better understanding under which circumstances the proposed method works well. In particular, it would be interesting how well the method works on a quite different search space, such as the one of MobileNet.